# Hierarchical Programmatic Option Framework

**Yu-An Lin**[*]  **Chen-Tao Lee**[*]  **Chih-Han Yang**[*]  **Guan-Ting Liu**[*]  **Shao-Hua Sun**
National Taiwan University
{b06204039, b06703027, b10902069, f07944014, shaohuas}@ntu.edu.tw

## Abstract

Deep reinforcement learning aims to learn deep neural network policies to solve large-scale decision-making problems. However, approximating policies using deep neural networks makes it difficult to interpret the learned decision-making process. To address this issue, prior works [10, 46, 74] proposed to use human-readable programs as policies to increase the interpretability of the decision-making pipeline. Nevertheless, programmatic policies generated by these methods struggle to effectively solve long and repetitive RL tasks and cannot generalize to even longer horizons during testing. To solve these problems, we propose the Hierarchical Programmatic Option framework (HIPO), which aims to solve long and repetitive RL problems with human-readable programs as options (low-level policies). Specifically, we propose a method that retrieves a set of effective, diverse, and compatible programs as options. Then, we learn a high-level policy to effectively reuse these programmatic options to solve reoccurring subtasks. Our proposed framework outperforms programmatic RL and deep RL baselines on various tasks. Ablation studies justify the effectiveness of our proposed search algorithm for retrieving a set of programmatic options.

## 1   Introduction

Deep reinforcement learning (deep RL) has recently achieved tremendous success in various domains, such as controlling robots [26, 31], playing strategy board games [67, 68], and mastering video games [78, 82]. However, neural network policies learned by deep RL methods are not human-interpretable, and their black-box nature poses challenges in scrutinizing model decisions and establishing user trust [45, 64]. Moreover, deep RL policies often suffer from overfitting and struggle to generalize to novel scenarios [17, 84], limiting their applicability in the context of most real-world applications.

To address these issues, programmatic RL frameworks [10, 46, 74] were proposed to represent policies as programs that detail task-solving procedures in a formal programming language. Particularly, Trivedi et al. [74] and  Liu et al. [46] synthesize programs from continuous latent spaces, whereas Carvalho et al. [10] search programs directly from discrete programmatic spaces. Such *program policies* are human-readable and demonstrate significantly improved zero-shot generalizability from smaller state spaces to larger ones.

Despite encouraging results, prior programmatic RL frameworks are limited to generating concise programs that can only tackle short-horizon tasks. Particularly for long-horizon RL problems like robotic manipulation [2, 8] and autonomous driving [55, 58], these tasks consist of reoccurring subtasks and sparse rewards, necessitating a significant number of actions to be fully resolved. Therefore, the RL agent needs to learn a diverse and reusable set of skills to solve such tasks effectively.

---

[*]Equal contribution.    Correspondence to: Shao-Hua Sun <shaohuas@ntu.edu.tw>

38th Conference on Neural Information Processing Systems (NeurIPS 2024).

Aiming to solve long and repetitive tasks with better policy interpretability, we borrow ideas from the option frameworks [4, 51, 66, 72] and latent-space-based programmatic RL frameworks [46, 74]. With that in mind, we propose the **Hi**erarchical **P**rogrammatic **O**ption framework (HIPO), which utilizes a set of programs with diverse skills as options (programmatic options) and learn a high-level policy to determine which programmatic option to be used based on the current state and the current option. By switching between these programmatic options, HIPO can reuse the skills encapsulated in these options to tackle long-horizon tasks with an arbitrary number of repeating subroutines.

Our framework contains three stages. (1) **Constructing a program embedding space**: To establish a program embedding space that smoothly and continuously parameterizes programs with diverse behaviors, we adopt the method proposed by Trivedi et al. [74]. (2) **Retrieving a diverse set of effective and reusable programmatic options**: We introduce a searching algorithm to retrieve a set of programmatic options from the learned program embedding space. Each programmatic option can be executed in the MDP and achieve satisfactory performance; more importantly, these programs are compatible and can be sequentially executed in any order. (3) **Learning a high-level policy**: To alter between a set of programmatic options, the high-level policy represented by neural networks takes the current environment state and the current programmatic option as input to predict the next programmatic option. This high-level policy can be learned using RL algorithms with the goal of maximizing the task return from the MDP.

To evaluate our proposed HIPO framework, we adopt the Karel domain [56], which characterizes an agent that navigates a grid world and interacts with objects. HIPO outperforms prior programmatic reinforcement learning and deep RL baselines on existing benchmarks [46, 74]. To further evaluate the performance and generalization ability to even longer horizons, we design a new set of tasks consisting of an arbitrary number of subtasks. Our framework shows better generalization in testing environments of different lengths. Ablation studies are also conducted to demonstrate the effectiveness of the proposed programmatic options retrieving process.

## 2    Related work

**Program synthesis.** Program synthesis techniques revolve around program generation to convert given inputs into desired outputs. These methods have demonstrated notable successes across diverse domains such as array and tensor manipulation [5, 22], string transformation [20, 29, 85], generating computer commands [44] and code [11, 42], graphics and 3D shape modeling [48, 73, 81], and describing agent behaviors [9, 13, 43, 69, 70]. Most program synthesis methods focus on task specifications such as input/output pairs, demonstrations, or language descriptions; in contrast, this work aims to synthesize human-readable programs as options to solve reinforcement learning tasks.

**Programmatic reinforcement learning.** Programmatic reinforcement learning methods [16, 80] explore structured representations for representing RL policies, including decision trees [7, 35], state machines [32], symbolic expressions [19, 33, 39, 49, 50, 83], and programs [1, 75, 76]. Liu et al. [46], Medeiros et al. [52], Moraes and Lelis [53], Trivedi et al. [74], and Carvalho et al. [10] attempted to produce policies described by domain-specific language programs to solve simple RL tasks. We aim to take a step toward addressing complex, long-horizon, repetitive tasks.

**Hierarchical reinforcement learning.** Hierarchical Reinforcement Learning (HRL) frameworks [4, 6, 40, 72, 77] focus on learning and operating across different levels of temporal abstraction, enhancing the efficiency of learning and exploration, particularly in sparse-reward environments. In this work, our proposed HIPO shares the same spirit with HRL frameworks [18, 21, 24, 25, 62, 63] that learn reusable skills as options. Instead of learning uninterpretable options as low-level policies, our framework aims to retrieve reusable and interpretable programs as options.

## 3    Problem formulation

Our goal is to devise a framework that generates a set of programs as options (low-level policies) and integrates them with high-level policies to tackle complex, long-term tasks defined by Markov Decision Processes (MDPs). To this end, we first synthesize a set of task-solving, diverse, compatible programs, then train a high-level policy to iteratively select and execute programs.

**Domain specific language.** This work adopts the domain-specific language (DSL) of the Karel domain [9, 13, 74], as illustrated in Figure 1. This DSL describes the control flows as well as the perception and actions of the Karel agent. Actions including `move`, `turnRight`, and `putMarker` define how the agent can interact with the environment. Perceptions, such as `frontIsClear` and `markerPresent`, formulate how the agent observes the environment. Control flows, *e.g.*, `if`, `else`, `while`, enable representing divergent and repetitive behaviors. Furthermore, Boolean and logical operators like `and`, `or`, and `not` allow for composing more intricate conditions. This work uses programs structured in this DSL to construct programmatic options.

**Markov Decision Process (MDP).** The tasks considered in this work can be formulated as finite-horizon discounted Markov Decision Processes (MDPs). The performance of HIPO is evaluated based on the execution traces of a series of programs (programmatic options) selected by its high-level policy. The rollout of a program $\rho$ consists of a $T$-step sequence of state-action pairs $\{(s_t, a_t)\}_{t=1, \dots, T}$ obtained from a program executor $\text{EXEC}(\cdot)$ that executes program $\rho$ to interact with an environment, resulting in the discounted return $\sum_{t=0}^{T} \gamma^t (r_t)$, where $r_t = \mathcal{R}(s_t, a_t)$ denotes the reward function. We aim to maximize the total rewards by executing a series of programs following the high-level policy.

| |
|---|
| Program $\rho \coloneqq$ DEF run m( $s$ m) |
| Repetition $n \coloneqq$ Number of repetitions |
| Perception $h \coloneqq$ frontIsClear \| leftIsClear \| rightIsClear \| markerPresent \| noMarkerPresent |
| Condition $b \coloneqq$ perception h \| not perception h |
| Action $a \coloneqq$ move \| turnLeft \| turnRight \| putMarker \| pickMarker |
| Statement $s \coloneqq$ while c( $b$ c) w( $s$ w) \| $s_1; s_2$ \| $a$ \| repeat R=$n$ r( $s$ r) \| if c( $b$ c) i( $s$ i) \| ifelse c( $b$ c) i( $s_1$ i) else e( $s_2$ e) |

Figure 1: **Karel domain-specific language (DSL)**, designed for describing the Karel agent's behaviors.

**Hierarchical Programmatic Option Framework.** The proposed hierarchical framework consists of a set of programmatic options $M = \{m_k\}_{k=1, \dots, |M|}$ as low-level policies and a high-level policy $f_\phi$ that sequentially chooses one option at a time. Each option $m_i$ encapsulates a human-readable program $\rho_{m_i}$ that will be executed if selected by the high-level policy $f_\phi$. On the other hand, the high-level policy $f_\phi(m^i, s^i_{T^i})$ outputs the probability distribution over all programmatic options $M$, given the last selected programmatic option $m^i$ at timestep $T^i$ and the current MDP state $s^i_{T^i}$. If the next option $m^{i+1}$ sampled from the distribution is the termination mode $m_{\text{term}}$, the rollout will be terminated. Otherwise, the corresponding programmatic option $\rho_{m^{i+1}}$ will be executed and generates a sequence of state-action pairs $\{(s^{i+1}_t, a^{i+1}_t)\}_{t=1, \dots, T^{i+1}}$ before the high-level policy $f_\phi$ selects the next programmatic option $m^{i+2}$.

# 4 Approach

We design a three-stage framework to search programmatic options and train a high-level policy represented by neural networks. The main goal is to maximize the return given a task described by an MDP. Firstly, as introduced in Section 4.1, we construct a program embedding space parameterizing programs smoothly and continuously with diverse behaviors. Then, Section 4.2 presents a method that retrieves a set of effective, diverse, and compatible programmatic options. Given retrieved programmatic options, Section 4.3 describes learning the high-level policy to determine probability distributions for options sampling. An overview of the proposed framework is illustrated in Figure 2.

## 4.1 Constructing program embedding space

We follow the approach and the program dataset specified in Trivedi et al. [74] to learn a program embedding space that smoothly and continuously parameterizes programs with diverse behaviors. The training objectives include a VAE loss and two losses that encourage learning a behaviorally smooth program embedding space. Once trained, we can use the learned decoder $p_\theta$ to map any program embedding $z$ to a program $\rho_z = p_\theta(z)$ consisting of a sequence of program tokens. Details on the program dataset generation and the encoder-decoder training can be found in Section E.1.1.

## 4.2 Retrieving programmatic options

With a program embedding space, we aim to retrieve a set of programs (programmatic options) given a task. This set of programs should have the following properties.

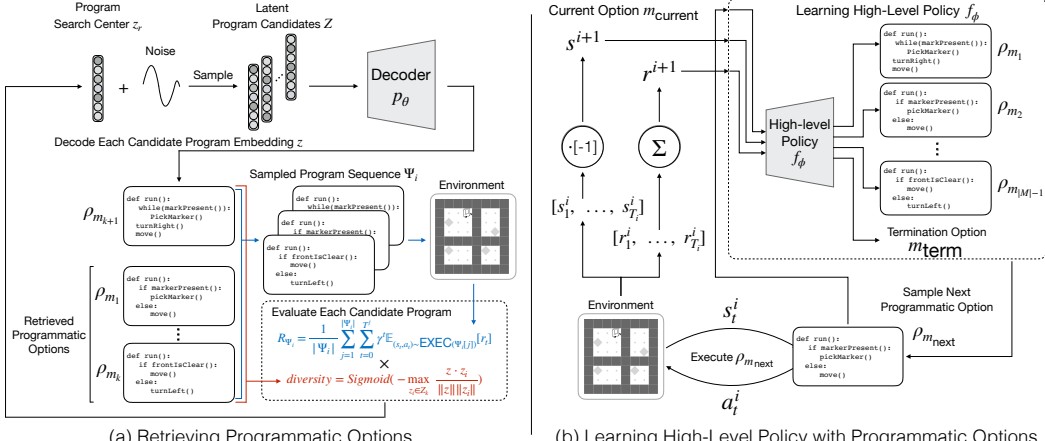

(a) Retrieving Programmatic Options        (b) Learning High-Level Policy with Programmatic Options

Figure 2: **Hierarchical Programmatic Option Framework. (a): Retrieving programmatic options.** After learning the program embedding space, we propose an advanced search scheme built upon the Cross-Entropy Method (CEM) to search programs $\rho_{m_1}, ..., \rho_{m_k}, \rho_{m_{k+1}}$ of different skills. While searching for the next program $\rho_{m_{k+1}}$, we consider its compatibility with predetermined programs $\rho_{m_1}, ..., \rho_{m_k}$ by randomly sampling a sequence of programs. We also consider the diversity among all programs using the *diversity multiplier*. **(b): Learning the high-level policy.** Given the current environment state $s$ and the current programmatic option $m_i$, the high-level policy outputs a probability distribution over all programmatic options, aiming to maximize the total accumulative reward from the environment.

- **Effectiveness**: Each program can solve the task to some extent.
- **Diversity**: The more behaviorally diverse the programs are, the richer behaviors can be captured.
- **Compatibility**: Sequentially executing some programs with specific orders can potentially lead to improved task performance.

### 4.2.1 Retrieving effective programs

To obtain a task-solving program, we can apply the Cross-Entropy Method [59], iteratively searching in a learned program embedding space [74] in the following procedure:

(1) Randomly initialize a program embedding vector $z_r$ as the search center.

(2) Add random noises to $z_r$ to generate a population of program embeddings $Z = \{z_i\}_{i=1,...,n}$, where $n$ denotes the population size.

(3) Evaluate every program embedding $z \in Z$ with the evaluation function $G$ to get a list of fitness score $[G(z_i)]_{i=1,...,n}$.

(4) Average the top k program embeddings in $Z$ according to fitness scores $[G(z_i)]_{i=1,...,n}$ and assign it to the search center $z_r$.

(5) Repeat (2) to (4) until the fitness score $G(z_r)$ of $z_r$ converges or the maximum number of steps is reached.

Since we aim to retrieve a set of effective programs, we can define the evaluation function as the program execution return of a decoded program embedding, *i.e.*, $G(z) = \sum_{t=0}^{T} \gamma^t \mathbb{E}_{(s_t,a_t) \sim \text{EXEC}(\rho_z)}[r_t]$. To retrieve a set of $|M|$ programmatic options for a high-level policy, we deploy this CEM search $N$ times, take $|M|$ best program embeddings, and obtain the decoded program set $\{\rho_{z_{r_i}} = p_\theta(z_{r_i})\}_{i=1,...,|M|}$. Please refer to Section A.1 for more details and the CEM search pseudo-code.

### 4.2.2 Retrieving effective, diverse programs

In our proposal, we retrieve a set of programmatic options for a high-level policy. Hence, the diversity of behaviors of the retrieved program set endow HIPO with versatility. However, by

running the CEM search for $|M|$ times, the obtained program set can have low diversity, making the multiskill-demanding tasks unsolvable.

To address this issue, we propose the *diversity multiplier* that accounts previous search results to encourage diversity among the retrieved programs. The evaluation of program employing the *diversity multiplier* is illustrated in Figure 2. Specifically, during the $(k + 1)$st CEM search, each program embedding $z$ is evaluated by $G(z, Z_k) = (\sum_{t=0}^{T} \gamma^t \mathbb{E}_{(s_t,a_t) \sim \text{EXEC}(\rho_z)}[r_t]) \cdot diversity(z, Z_k)$, where $diversity(z, Z_k)$ is the proposed *diversity multiplier* defined as $Sigmoid(-\max_{z_i \in Z_k} \frac{z \cdot z_i}{\|z\|\|z_i\|})$. Thus, the program execution return is scaled down by $diversity(z, Z_k)$ based on the maximum cosine similarity between $z$ and the retrieved program embeddings $Z_k = \{z_i\}_{i=1,\dots,k}$ from the previous $k$ CEM searches, diverging the current program embedding from previously retrieved programs.

To retrieve a set of $|M|$ programmatic options for our high-level policy, we deploy this CEM+diversity search $N$ times, take $|M|$ best program embeddings, and obtain the decoded program set. The procedure and the search trajectory visualization can be found in Section A.2.

### 4.2.3 Retrieving effective, diverse, compatible programs

The proposed HIPO executes a sequence of programmatic options determined by a high-level policy. Therefore, these programs shall be compatible with each other, *i.e.*, executing a program following the execution of other programs could improve task performance. Yet, CEM+diversity discussed in Section 4.2.2 searches every program independently.

Accounting for the compatibility among programs while searching, we propose an evaluation method, CEM+diversity+compatibility. To evaluate the program embedding $z$, we take the decoded program $\rho_z$ as the $(k + 1)$st option. Then, lists of programs $\Psi_{i,i=1,\dots,D}$ are sampled with replacements from determined $k$ options and the $(k + 1)$st option. Each program list $\Psi_i$ contains at least one $(k + 1)$st option to consider the compatibility between the $(k + 1)$st and previously determined $k$ options. The return is computed by sequentially executing these $D$ sequences of programs and multiply the result with the *diversity multiplier* proposed in Section 4.2.2. As the result, the evaluation function is $G(z, Z_k) = \frac{1}{D} \sum_{i=1}^{D} R_{\Psi_i} \cdot diversity(z, Z_k)$, where $R_{\Psi_i}$ is the normalized reward obtained from executing all programs in the specified program sequence $\Psi_i$:

$$R_{\Psi_i} = \frac{1}{|\Psi_i|} \sum_{j=1}^{|\Psi_i|} \sum_{t=0}^{T^j} \gamma^t \mathbb{E}_{(s_t,a_t) \sim \text{EXEC}(\Psi_i[j])}[r_t], \tag{1}$$

with $|\Psi_i|$ being the number of programs in the program sequence $\Psi_i$, $\Psi_i[j]$ being the $j$-th program, and $\gamma$ as the discount factor.

The search method with the specified evaluation is deployed $|M|$ times to obtain a set of programs that are effective, diverse, and compatible with each other, adopted by the high-level policy as programmatic options. Please refer to Section A.3 for more details and the thorough procedure.

### 4.3 Learning high-level policy with programmatic options

Given a set of programmatic options $M = \{m_k\}_{k=1,\dots,|M|}$, we formulate learning a high-level policy $f_\phi$ represented by neural networks, as a reinforcement learning problem aiming to maximize the task return. At the $i$-th high-level step, given the latest selected programmatic option $m^i$ and the current environment state $s$, the high-level policy $f_\phi(m^i, s)$ outputs the probability distribution of programmatic options for the next option $m^{i+1} \in m_{\text{term}} \cup \{m_k\}_{k=1,\dots,|M|}$, where $m_{\text{term}}$ denotes the termination option that ends the episode once selected. Otherwise, the corresponding program $\rho_{m^{i+1}}$ is executed, yielding the next state $s_{T^{i+1}}^{i+1}$ and the cumulative reward $r^{i+1} = \sum_{t=1}^{T^{i+1}} r_t^{i+1}$. Note that the last state $s_{T^{i+1}}^{i+1}$ of the state sequence is returned by $\text{EXEC}(\rho_{m^{i+1}})$, with $T^{i+1}$ denoting the horizon of the $i + 1$-th program execution. Also, the cumulative reward $r^{i+1}$ obtained within a single program execution is not discounted.

Note that a single program execution $\text{EXEC}(\rho)$ will terminate after complete execution or the number of function calls emitted during $\text{EXEC}(\rho)$ reaches 220, which aligns to the setting in Trivedi et al.

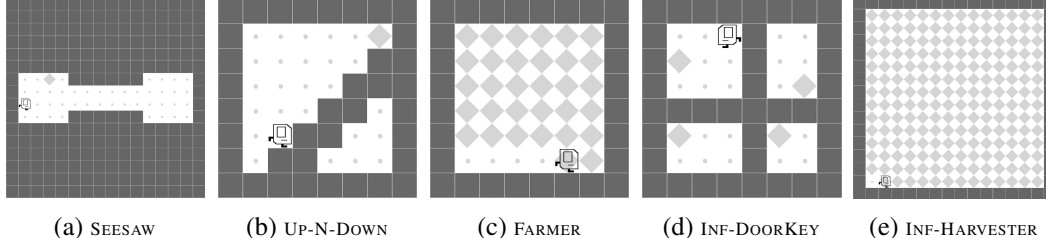

| (a) SEESAW | (b) UP-N-DOWN | (c) FARMER | (d) INF-DOORKEY | (e) INF-HARVESTER |

Figure 3: **KAREL-LONG problem set**: This work introduces a new set of tasks in the Karel domain. These tasks necessitate learning diverse, repetitive, and task-specific skills. For example, in our designed INF-HARVESTER, the agent needs to traverse the whole map and pick nearly 400 markers to solve the tasks since the environment randomly generates markers; in contrast, the HARVESTER from the KAREL problem set [74] can be solved by picking merely 36 markers.

[74]. This iterative process stops once the termination option is sampled or the maximum number of the option-selection steps is reached. Please refer to Section E.1.3 for training details.

To further enhance the explainability of the high-level policy, we apply the approach proposed by Koul et al. [36] to extract the state machine structure from the learned high-level policy. Combining the retrieved set of programmatic options and the extracted state machine structure, our framework is capable of solving long-horizon tasks while being self-explanatory. Examples of extracted state machines are illustrated in Section D.

## 5 Experiments

We aim to answer the following questions with the experiments and ablation studies. (1) Can our proposed *diversity multiplier* introduced in Section 4.2.2 enhance CEM and yield programs with improved performance? (2) Can our proposed CEM+diversity+compatibility introduced in Section 4.2.3 retrieve a set of programs that are diverse yet compatible with each other? (3) Can the proposed framework outperforms existing methods on long-horizon tasks?

Please refer to Section E for hyperparameter settings for the following experiments.

### 5.1 Karel problem sets

To this end, we consider the Karel domain [56], which is widely adopted in program synthesis [9, 13, 65, 70] and programmatic reinforcement learning [46, 74]. Specifically, we utilize the KAREL problem set [74] and the KAREL-HARD problem set [46]. The KAREL problem set includes six basic tasks, each of which can be solved by a short program (less than 45 tokens), with a horizon shorter than 200 steps per episode. On the other hand, the four tasks introduced in the KAREL-HARD problem require longer, more complex programs (*i.e.*, 45 to 120 tokens) in longer execution horizons (*i.e.*, up to 500 actions). Details on two problem sets is provided in Section F and Section G.

**KAREL-LONG problem set.** Since most of the tasks in the KAREL and KAREL-HARD problem sets are short-horizon tasks (*i.e.*, can be finished in less than 500 timesteps), they are not suitable for evaluating long-horizon task-solving ability (*i.e.*, tasks requiring more than 3000 timesteps to finish). Hence, we introduce a newly designed KAREL-LONG problem set as the benchmark to evaluate the capability of HIPO.

As illustrated in Figure 3, the tasks require the agent to fulfill extra constraints (*e.g.*, not placing multiple markers on the same spot in FARMER, receiving penalties imposed for not moving along the stairs in UP-N-DOWN) and conduct extended exploration (*e.g.*, repetitively locating and collecting markers in SEESAW, INF-DOORKEY, and INF-HARVESTER). More details on the KAREL-LONG tasks can be found in Section H.

### 5.2 Cross-entropy method with diversity multiplier

We aim to investigate whether our proposed *diversity multiplier* can enhance CEM and yield programs with improved performance. To this end, for each KAREL or KAREL-HARD task, we use CEM and

Table 1: **Evaluation on KAREL and KAREL-HARD tasks.** Mean return and standard deviation of all methods across the KAREL and KAREL-HARD problem set, evaluated over five random seeds. CEM+diversity outperforms CEM with significantly smaller standard deviations across 8 out of 10 tasks, highlighting the effectiveness and stability of CEM+diversity. In addition, HIPO outperforms LEAPS and HPRL on 8 out of 10 tasks.

| Method | FOUR CORNER | TOP OFF | CLEAN HOUSE | STAIR CLIMBER | HARVESTER | MAZE | DOOR KEY | ONE STROKE | SEEDER | SNAKE |
|---|---|---|---|---|---|---|---|---|---|---|
| CEM | $0.45 \pm 0.40$ | $0.81 \pm 0.07$ | $0.18 \pm 0.14$ | $\mathbf{1.00} \pm 0.00$ | $0.45 \pm 0.28$ | $\mathbf{1.00} \pm 0.00$ | $0.50 \pm 0.00$ | $0.65 \pm 0.19$ | $0.51 \pm 0.21$ | $0.21 \pm 0.15$ |
| CEM+diversity | $\mathbf{1.00} \pm 0.00$ | $\mathbf{1.00} \pm 0.00$ | $0.37 \pm 0.06$ | $\mathbf{1.00} \pm 0.00$ | $0.80 \pm 0.07$ | $\mathbf{1.00} \pm 0.00$ | $0.50 \pm 0.00$ | $0.62 \pm 0.01$ | $0.69 \pm 0.07$ | $0.36 \pm 0.02$ |
| DRL | $0.29 \pm 0.05$ | $0.32 \pm 0.07$ | $0.00 \pm 0.00$ | $\mathbf{1.00} \pm 0.00$ | $0.90 \pm 0.10$ | $\mathbf{1.00} \pm 0.00$ | $0.48 \pm 0.03$ | $\mathbf{0.89} \pm 0.04$ | $0.96 \pm 0.02$ | $\mathbf{0.67} \pm 0.17$ |
| LEAPS | $0.45 \pm 0.40$ | $0.81 \pm 0.07$ | $0.18 \pm 0.14$ | $\mathbf{1.00} \pm 0.00$ | $0.45 \pm 0.28$ | $\mathbf{1.00} \pm 0.00$ | $0.50 \pm 0.00$ | $0.65 \pm 0.19$ | $0.51 \pm 0.21$ | $0.21 \pm 0.15$ |
| HPRL | $\mathbf{1.00} \pm 0.00$ | $\mathbf{1.00} \pm 0.00$ | $\mathbf{1.00} \pm 0.00$ | $\mathbf{1.00} \pm 0.00$ | $\mathbf{1.00} \pm 0.00$ | $\mathbf{1.00} \pm 0.00$ | $0.50 \pm 0.00$ | $0.80 \pm 0.02$ | $0.58 \pm 0.07$ | $0.28 \pm 0.11$ |
| HIPO (Ours) | $\mathbf{1.00} \pm 0.00$ | $\mathbf{1.00} \pm 0.00$ | $\mathbf{1.00} \pm 0.00$ | $\mathbf{1.00} \pm 0.00$ | $\mathbf{1.00} \pm 0.00$ | $\mathbf{1.00} \pm 0.00$ | $\mathbf{1.00} \pm 0.00$ | $0.62 \pm 0.01$ | $\mathbf{0.97} \pm 0.02$ | $0.36 \pm 0.02$ |

CEM+diversity to find 10 programs. Then, for each task, we evaluate all the programs and report the best performance in Table 1. The results suggest that our proposed CEM+diversity achieves better performance on most of the tasks, highlighting the improved search quality induced by covering wider regions in the search space with the *diversity multiplier*. Visualized search trajectories of CEM+diversity can be found in Section A.2.

## 5.3 Ablation study

We propose CEM+diversity+compatibility to retrieve a set of effective, diverse, compatible programmatic options for our high-level policy. In this section, we compare a variety of implementations regarding the diversity and the compatibility of programs.

- **CEM** $\times |M|$: Conduct the CEM search described in Section 4.2.1 $|M|$ times and take the resulting $|M|$ programs as the set of programmatic options.

- **CEM+diversity top** $k$**,** $k = |M|$: Conduct the CEM search with the *diversity multiplier* described in Section 4.2.2 $N = 10$ times and take the top $|M|$ results as the set of programmatic options.

- **CEM+diversity** $\times |M|$: Conduct the CEM search with the *diversity multiplier* described in Section 4.2.2 $N = 10$ times and select the best program as the $i^{th}$ option. Repeat this process $|M|$ times to extract $|M|$ programs as the set of programmatic options.

- **CEM+compatibility** $\times |M|$: Conduct the CEM search by executing programs in the specified program sequence $\Psi_i$ described in Section 4.2.3, excluding the *diversity multiplier*. Iteratively perform this search $|M|$ times and take the resulting $|M|$ programs as the set of programmatic options.

- **HIPO (ours)**: Conduct CEM+diversity+compatibility (*i.e.*, CEM with the *diversity multiplier* and $R_\Psi$ as described in Section 4.2.3) for $N = 10$ times and select the best result as the $i^{th}$ option. Repeat the above process $|M|$ times and take all $|M|$ results as the set of programmatic options.

The number of programmatic options $|M|$ is 3 for SEESAW, UP-N-DOWN, and INF-HARVESTER and 5 for FARMER and INF-DOORKEY. We assess the quality of the retrieved programmatic options by evaluating the performance of the high-level policy learned with these option sets on the KAREL-LONG tasks. The results presented in Table 2 indicate that our proposed framework, HIPO, outperforms its variants that ignore diversity or compatibility among programmatic options on all the tasks. This justifies our proposed CEM+diversity+compatibility method for retrieving a set of effective, diverse, compatible programs as options for the high-level policy.

## 5.4 Comparing with deep RL and programmatic RL Methods

We compare our proposed framework and its variant to state-of-the-art deep RL and programmatic RL methods on the KAREL-LONG tasks.

- **Random transition** uses the same set of programmatic options as our method but with a random high-level policy (*i.e.*, uniformly randomly select the next option at each step). The performance of this method examines the necessity of learning a high-level policy.

Table 2: **KAREL-LONG performance.** Mean return and standard deviation of all methods across the KAREL-LONG problem set, evaluated over five random seeds. Our proposed framework achieves the best mean reward across most of the tasks by learning a high-level policy with a set of effective, diverse, and compatible programs.

| Method | SEESAW | UP-N-DOWN | FARMER | INF-DOORKEY | INF-HARVESTER |
|---|---|---|---|---|---|
| CEM $\times|M|$ | $0.06 \pm 0.10$ | $0.39 \pm 0.36$ | $0.03 \pm 0.00$ | $0.11 \pm 0.14$ | $0.41 \pm 0.17$ |
| CEM+diversity top $k$, $k = |M|$ | $0.15 \pm 0.21$ | $0.25 \pm 0.35$ | $0.03 \pm 0.00$ | $0.13 \pm 0.16$ | $0.42 \pm 0.19$ |
| CEM+diversity $\times|M|$ | $0.28 \pm 0.23$ | $0.58 \pm 0.31$ | $0.03 \pm 0.00$ | $0.36 \pm 0.26$ | $0.47 \pm 0.23$ |
| CEM+compatibility $\times|M|$ | $0.23 \pm 0.32$ | $0.43 \pm 0.34$ | $0.14 \pm 0.22$ | $0.57 \pm 0.3$ | $0.66 \pm 0.08$ |
| Random Transition | $0.01 \pm 0.00$ | $0.02 \pm 0.01$ | $0.01 \pm 0.00$ | $0.01 \pm 0.01$ | $0.15 \pm 0.04$ |
| PAO | $0.01 \pm 0.01$ | $0.00 \pm 0.00$ | $0.43 \pm 0.23$ | $0.34 \pm 0.45$ | $0.60 \pm 0.04$ |
| DRL | $0.00 \pm 0.01$ | $0.00 \pm 0.00$ | $0.38 \pm 0.25$ | $0.17 \pm 0.36$ | $0.74 \pm 0.05$ |
| Option-Critic | $0.00 \pm 0.00$ | $0.00 \pm 0.00$ | $0.00 \pm 0.00$ | $0.00 \pm 0.00$ | $0.47 \pm 0.01$ |
| LEAPS | $0.01 \pm 0.01$ | $0.02 \pm 0.01$ | $0.03 \pm 0.00$ | $0.01 \pm 0.00$ | $0.12 \pm 0.00$ |
| HPRL | $0.00 \pm 0.00$ | $0.00 \pm 0.00$ | $0.01 \pm 0.00$ | $0.00 \pm 0.00$ | $0.45 \pm 0.03$ |
| HC | $0.22 \pm 0.08$ | $0.31 \pm 0.38$ | $0.19 \pm 0.03$ | $0.14 \pm 0.16$ | $\mathbf{0.88} \pm 0.00$ |
| HIPO (Ours) | $\mathbf{0.53} \pm 0.10$ | $\mathbf{0.76} \pm 0.02$ | $\mathbf{0.62} \pm 0.02$ | $\mathbf{0.66} \pm 0.07$ | $0.79 \pm 0.02$ |

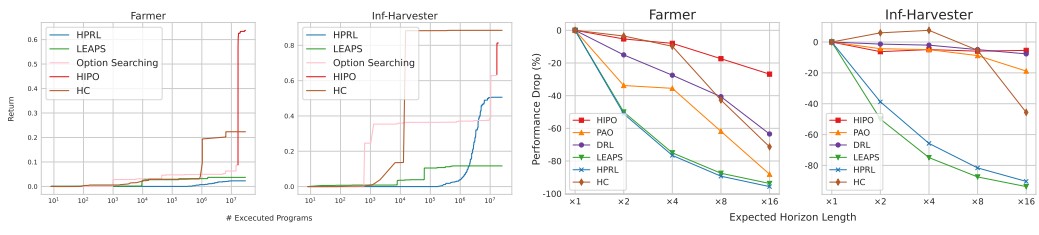

(a) Program Sample Efficiency     (b) Inductive Generalization

Figure 4: (a) **Program sample efficiency.** The training curves of HIPO and other programmatic RL approaches, where the x-axis is the total number of executed programs for interacting with the environment, and the y-axis is the maximum validation return. This demonstrates that our proposed framework has better program sample efficiency and converges to better performance. (b) **Inductive generalization performance.** We evaluate and report the performance drop in the testing environments with an extended horizon, where the x-axis is the extended horizon length compared to the horizon of the training environments, and the y-axis is the performance drop in percentage. Our proposed framework can inductively generalize to longer horizons without any fine-tuning.

- **Primitive actions as options (PAO)** learns a high-level policy similar to HIPO, which takes the current option and environment state as input and predicts the next option. However, it utilizes primitive actions (*e.g.*, move, pickMarker) as options. This baseline highlights the necessity of retrieving programs with higher-level behaviors as options.

- **DRL** represents a policy as a neural network and is learned using PPO [61]. The policy takes raw states (*i.e.*, Karel grids) as input and predicts the probability distribution over the set of primitive actions, (*e.g.*, move, pickMarker).

- **Option-Critic** represents a policy that both high-level and low-level policies are neural networks and is learned using the option-critic architecture [4]. The policy takes raw states (*i.e.*, Karel grids) as input, and each option predicts the probability distribution over the set of primitive actions (*e.g.*, move, pickMarker).

- **Learning Embeddings for Latent Program Synthesis (LEAPS)** [74] searches for a single task-solving program using the vanilla CEM in a learned program embedding space.

- **Hierarchical Programmatic Reinforcement Learning (HPRL)** [46] learns a meta-policy, whose action space is a learned program embedding space, to compose a series of programs as the policy.

- **Hill Climbing (HC)** [10] is a stochastic search technique that operates directly within the program space. The process begins by randomly modifying portions of the current program to generate a set of neighboring candidates. The program that performs best among these neighbors is selected as the next candidate for exploration.

As Table 1 shows, HIPO outperforms LEAPS and HPRL on 8 out of 10 tasks from the KAREL and KAREL-HARD tasks, indicating that the retrieved programs are truly effective at solving short horizon

tasks (*i.e.*, less than 500 actions). For long-horizon tasks that require more than 3000 actions to solve, Table 2 shows that HIPO excels on four tasks, with better performance on FARMER and particular prowess in SEESAW, UP-N-DOWN, and INF-DOORKEY.

Two of these tasks require distinct skills (*e.g.*, pick and put markers in FARMER; go up and downstairs in UP-N-DOWN) and the capability to persistently execute one skill for an extended period before transitioning to another. HIPO adeptly addresses this challenge due to the consideration of diversity while seeking programmatic options, ensuring the acquisition of both skills concurrently.

Unlike the other tasks, SEESAW and INF-DOORKEY require an extended traverse to collect markers, leading to a sparser reward distribution. During the searching phase of programmatic options, emphasizing compatibility enables HIPO to secure a set of mutually compatible options that work together effectively to accomplish the extended traversal.

Retrieved programs are provided in Appendix (Figure 22, Figure 23, Figure 24, and Figure 25). Experimental results on programmatic policy baselines over 32 seeds are provided in Table 3.

### 5.5 Program sample efficiency

To accurately evaluate the sample efficiency of programmatic RL methods, we propose the *program sample efficiency* metric, measuring the total number of program executions required to learn a program policy. We report the program sample efficiency of LEAPS, HPRL, HC, and HIPO on FARMER and INF-HARVESTER in Figure 4a. As the results show, HIPO demonstrates program sample efficiency than LEAPS and HPRL, indicating that our framework requires fewer program interactions with the environment and lower computational costs compared to existing latent-space-based programmatic RL frameworks. More details and the action sample efficiency can be found in Section B and Figure 8.

### 5.6 Inductive generalization

We aim to compare the inductive generalization ability among all the methods, generalizing to out-of-distributionally (*i.e.*, unseen during training) long task instances [32]. To this end, we increase the expected horizons of FARMER and INF-HARVESTER by $2\times$, $4\times$, $8\times$, and $16\times$. Then, we report the performance drop compared to the original task performance of selected baselines in Figure 4b. More details on extending task horizons are provided in Section C.

The results show that HIPO suffers a fewer decline in performance in testing environments with significantly extended horizons than LEAPS and HPRL, suggesting that HIPO exhibits better inductive generalization in these tasks. The longest execution of HIPO ran up to 48k environment steps.

### 5.7 Interpretability

In the proposed framework, we retrieve a set of programmatic options that can be reused by the high-level policy to solve long and repetitive tasks. For example, in INF-DOORKEY (detailed in Section H), the retrieved programmatic options are presented in Figure 24. Based on these programmatic options, the high-level policy can reuse them to guide the agent to traverse all four chambers with the following sequence of programmatic options: $\tau_m = \{m_5, m_3, m_5, m_3, m_1, m_4, m_4, m_2, m_5, m_4, m_5\}$.

INF-DOORKEY requires three different skills to solve: picking a marker to open the door in some chambers, placing a marker to open the door in some other chambers, and navigating between the chambers. Specifically, the programs provided in Figure 24 show that Option 1 and Option 5 fulfill the first skill, while Option 3, Option 4, and Option 5 meet the requirements for the second, and Option 2 satisfy the third. Along with the observation, we could fully interpret the agent policy with the control flows, perceptions, and actions specified in the program. On the other hand, the high-level policy is a neural network by itself, making it hard to interpret the option transition dynamic learned by the high-level policy.

To improve the interpretability of our high-level policy, we extract state machine structures from the high-level policy to visualize the option transition dynamics. More details and examples of extracted state machines can be found in Section D.

# 6 Conclusion

This work aims to construct reinforcement learning policies that are human-interpretable and generalizable, bridging from hierarchical reinforcement learning to programmatic options. Consequently, we propose the Hierarchical Programmatic Option framework (HIPO) to represent complex behaviors and address long-horizon tasks. Specifically, we introduce a method that can retrieve a set of effective, diverse, compatible programs by modifying the Cross Entropy Method (CEM). Following this, these programs are applied as options by the high-level policy learned with reinforcement learning. To evaluate HIPO's ability in extended horizons, we design a set of tasks that require thousands of steps in the Karel domain. Our framework HIPO outperforms various deep RL and programmatic RL methods on various tasks. In addition, HIPO demonstrates good performance in inductive generalization to even longer horizons without fine-tuning. Last but not least, extensive ablation studies justify the effectiveness of our proposed search algorithm in retrieving programmatic options.

## Acknowledgement

Shao-Hua Sun was supported by the Yushan Fellow Program by the Ministry of Education, Taiwan.

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

**Appendix**

# Table of Contents

# A  Cross entropy method details

## A.1  CEM

Figure 5 illustrates the workflow of the cross entropy method (CEM). The corresponding pseudo-code is available in Algorithm 1.

Hyperparameters list:

- Population size $n$: 64
- Standard Deviation of Noise $\sigma$: 0.5
- Percent of the Population Elites $e$: 0.05
- Exponential $\sigma$ decay: True
- Maximum Iteration $N_s$: 1000

## A.2  CEM+diversity

The procedure of running CEM+diversity N times is as follows:

(1) Search the $1st$ program embedding $z_1$ by $CEM(G, g = (Z_k : \{\}))$

(2) Search the $2nd$ program embedding $z_2$ by $CEM(G, g = (Z_k : \{z_1\}))$
    ...

(N) Search the $Nth$ program embedding $z_N$ by $CEM(G, g = (Z_k : \{z_1, ..., z_{N-1}\}))$

$Z_k$ is the set of retrieved program embeddings $\{z_i\}_{i=1,...,k-1}$ from the previous $(k-1)$ CEM searches. The evaluation function is $G(z, Z_k) = (\sum_{t=0}^{T} \gamma^t \mathbb{E}_{(s_t, a_t) \sim \text{EXEC}(\rho_z)}[r_t]) \cdot diversity(z, Z_k)$, where $diversity(z, Z_k) = Sigmoid(-\max_{z_i \in Z_k} \frac{z \cdot z_i}{\|z\|\|z_i\|})$. Searching trajectories shown in Figure 6 exemplifies the influence of the diversity factor.

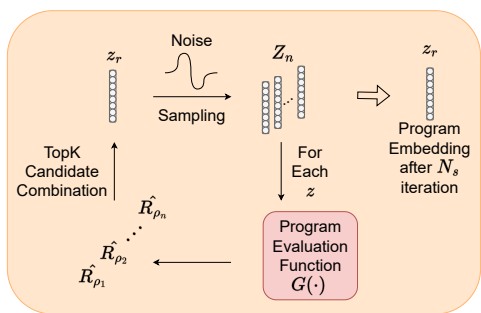

Figure 5: Using the Cross-Entropy Method to search for a program with high execution reward in the learned program embedding space.

**Algorithm 1** Cross Entropy Method

---

1: Input: Evaluation Function $G$, Function Input $g$, Maximum Iteration $N_s$, Population Size $n$, Standard Deviation of Noise $\sigma$, Percent of the Population Elites $e$.
2: Latent Program Search Center $z_r \leftarrow [z_0, z_1, ..., z_i, ..., z_{255}], z_i \sim \mathcal{N}(0, 1)$
3: $step \leftarrow 0$
4: **while** $step < N_s$ **do**
5:     Candidate Latent Programs $Z \leftarrow [\,]$
6:     Fitness Scores $L_G \leftarrow [\,]$
7:     **for** $i \leftarrow 1$ to $n$ **do**
8:         $\varepsilon \leftarrow [\varepsilon_0, \varepsilon_1, ..., \varepsilon_i, ..., \varepsilon_{255}], \varepsilon_i \sim \mathcal{N}(0, \sigma)$
9:         $Z$.append($z_r + \varepsilon$)
10:        $L_G$.append($G((z_r + \varepsilon), g)$)
11:     **end for**
12:     Elite Latent Programs $Z^{kl} \leftarrow$ Latent Programs in top $e$ percent of $Z$ ranked by $L_G$.
13:     $z_r \leftarrow mean(Z^{kl})$
14:     $step \leftarrow step + 1$
15: **end while**

---

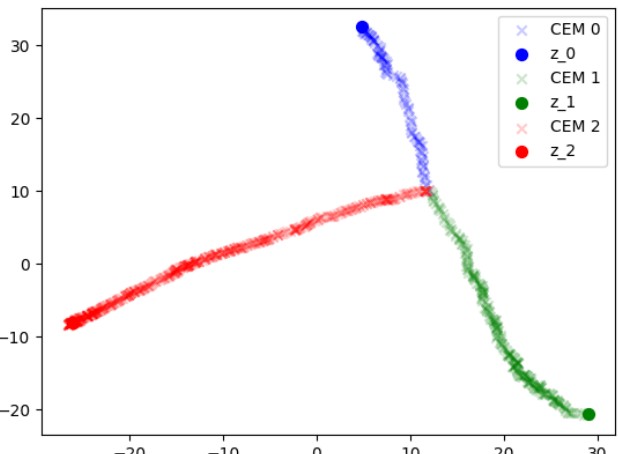

Figure 6: **CEM+diversity searching trajectories.** A demonstration of 3 searching trajectories of the CEM+diversity procedure in the latent space. The CEM-acquired program embeddings are reduced into 2-dimensional representation with PCA. Given the diversity factor, the $2nd$ CEM-search exploration is ushered in the opposite direction of the searching trajectory of the $1st$ CEM-search, and the $3rd$ CEM-search trajectory is perpendicular to the $1st$ and $2nd$ searching paths.

### A.3 CEM+diversity+compatibility

#### A.3.1 Sample program sequence

In the subsequent section, we outline the procedure for sampling a program sequence $\Psi$ from $k$ previously determined programs $\rho_{i,i=1,...,k}$ during the search of the $(k+1)$st program $\rho_{k+1}$.

(1) Uniformly sample a program $\rho_j$ from all $k+1$ programs $\{\rho_1, ..., \rho_k, \rho_{k+1}\}$, and add $\rho_j$ to $\Psi$.

(2) Repeat (1) until the $(k+1)$st program $\rho_{k+1}$ is sampled.

(3) Uniformly sample a program $\rho_j$ from $\{\rho_1, ..., \rho_k, \rho_{k+1}, \rho_{\text{term}}\}$, where $\rho_{\text{term}}$ represents the termination program, then append $\rho_j$ to $\Psi$.

(4) Repeat (3) until $\rho_{\text{term}}$ is sampled.

(5) Return $\Psi$ with length greater or equal to $L_{min}$, where $L_{min}$ is a hyperparameter that indicates the minimum length of a sampled sequence. Re-sample $\Psi$ otherwise.

#### A.3.2 The score function

The score function $G(z, Z_k)$ for CEM+diversity+compatibility:

$$G(z, Z_k) = \frac{1}{D} \sum_{i=1}^{D} R_{\Psi_i} \cdot diversity(z, Z_k) \tag{2}$$

$$R_{\Psi_i} = \frac{1}{|\Psi_i|} \sum_{j=1}^{|\Psi_i|} \sum_{t=0}^{T^j} \gamma^t \mathbb{E}_{(s_t, a_t) \sim \text{EXEC}(\Psi_i[j])}[r_t] \tag{3}$$

$|\Psi_i|$ denotes the number of programs in the program list $\Psi_i$, $\Psi_i[j]$ represents the $j$-th program in the program list $\Psi_i$, and $\gamma$ is the discount factor.

#### A.3.3 CEM+diversity+compatibility procedure

The procedure of running CEM+diversity+compatibility $|M|$ times in order to retrieve $|M|$ programs:

(1) Retrieve $1st$ program $z_1$.

    a. Sample a program sequence $\Psi_{i=1}$ with $k = 0$

    b. Deploy CEM+diversity N times with $Z_k = \{\}$ to acquire N program embeddings.

    c. Select the program embedding with the highest score $G(z, Z_k = \{\})$ as $z_1$, among the N program embeddings.

(2) Retrieve $2nd$ program $z_2$.

    a. Sample a program sequence $\Psi_{i=1,2}$ with $k = 1$ previously determined program.

    b. Deploy CEM+diversity N times with $Z_k = \{z_1\}$, to acquire N program embeddings.

    c. Select the program embedding with the highest score $G(z, Z_k = \{z_1\})$ as $z_2$, among the N program embeddings.

    ...

($|M|$) Retrieve $|M|th$ program $z_{|M|}$.

    a. Sample a program sequence $\Psi_{i,i=1,...,2^{|M|-1}}$ with $k = |M| - 1$ previously determined programs.

    b. Deploy CEM+diversity N times with $Z_k = \{z_1, z_2, ..., z_{|M|-1}\}$, to acquire N program embeddings.

    c. Select the program embedding with the highest score $G(z, Z_k = \{z_1, z_2, ..., z_{|M|-1}\})$ as $z_{|M|}$, among the N program embeddings.

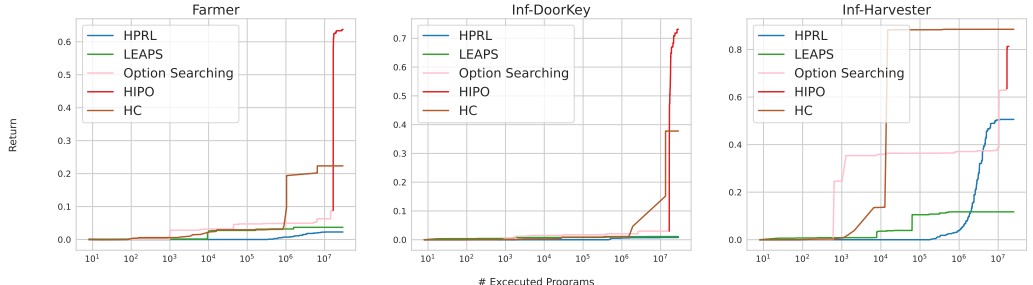

Figure 7: **Program sample efficiency.** Results of different programmatic RL approaches in FARMER, INF-DOORKEY, INF-HARVESTER.

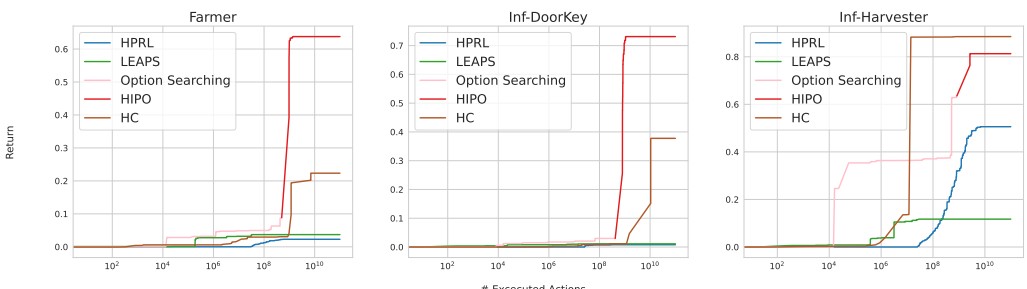

Figure 8: **Sample efficiency.** Results of different programmatic RL approaches in FARMER, INF-DOORKEY, INF-HARVESTER.

## B  Program sample efficiency

Generally speaking, programmatic RL approaches incorporate a three-step training procedure, in which a program will be first synthesized, exectued, then evaluated with respect to a given task. The three-step procedure is iteratively applied until the return converges or the maximum training step is reached. "Program Sample Efficiency"—referring to how effectively the method achieves a desired return using an efficient number of the three-step procedure—is crucial for analyzing and comparing the efficiency of our approaches against the baselines. As Figure 7 illustrates, HIPO (our method) obtains the best program sampling efficiency in FARMER and INF-DOORKEY. Details of the program sample efficiency calculation for each baseline are elaborated in the subsequent sections.

### B.1  HIPO

In the program searching phase of HIPO, up to 50 CEM-searches are conducted to fetch programmatic options. Each CEM-search involves a maximum of 1000 search iterations, in which the three-step procedure (synthesis, execution, and evaluation) is executed $n$ times, where $n$ is the population size of the CEM-search. In the first phase of the efficiency curve of our method (indicated by the yellow curve in Figure 7), the return is counted as the highest return obtained from a sequential execution of the programmatic options, based on the order indicated by the sampled random sequence.

Afterward, during the high-level policy training phase (indicated by the red curve in Figure 7), the three-step procedure is executed once per PPO training step. The return is recorded as the maximum validation return achieved in the HIPO manner, where the high-level policy continuously selects and deploys a program to solve a long-horizon task.

### B.2  LEAPS

In the program searching process of LEAPS, 216 CEM-searches are performed to obtain the targeted program, in accordance with the settings in [74]. In each CEM-search, up to 1000 iterations are performed, in which the three-step procedure is executed $n$ times, where $n$ is the population size of

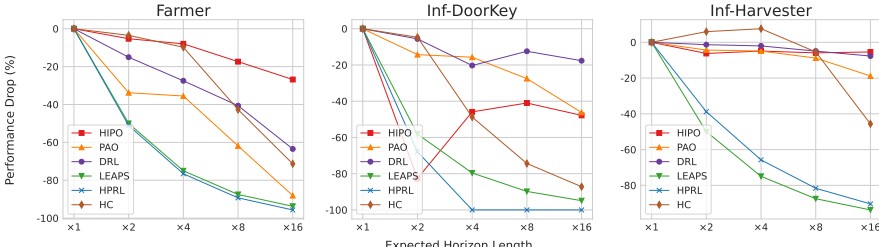

Figure 9: **Inductive generalization.** Experiment results on different baselines in FARMER, INF-DOORKEY, and INF-HARVESTER.

the CEM-search. The return for a certain number of executed programs is recorded as the maximum return obtained from executing the previously searched programs only.

### B.3 HPRL

During the meta-policy training process of HPRL, the three-step procedure is performed once per PPO training step. Therefore, with the exact experiment settings described in Section E.6, the three-step procedure would be executed 25M times at the end of the training process. The return for a certain number of executed programs is recorded as the maximum return achieved in the cascaded execution of 5 programs, decoded from the latent program embeddings output by the meta-policy.

### B.4 HC

In the program searching progress of HC, 250 neighbors are sampled and executed to obtain the targeted program for each iteration. Up to 1000000 evaluations would be performed in the searching process, with 16 parallel execution of a program involved in one evaluation. The return for a certain number of executed programs is recorded as the maximum return obtained from executing the previously searched programs only.

## C   Inductive generalization

To compare baselines with our proposal regarding inductive generalization, the expected horizon of the environment is scaled up by increasing the upper limit of the target for each KAREL-LONG task during the testing phase. For instance, in the FARMER task, the upper limit number is essentially the maximum iteration number of the filling-and-collecting rounds that we expect the agent to accomplish (please refer to Section H for more details). In the original task setting, the agent is tasked to continuously placing and picking markers in a total of 10 rounds of the filling-and-collecting process. In other words, all policies among baselines are trained to terminate after 10 placing-and-picking iterations. Nevertheless, the upper limit number is set to 20, 40, etc, in the testing environment regarding the inductive generalization ability of each policy.

Since most of the baselines obtain a poor peformance on SEESAW and UP-N-DOWN (*i.e.*, more than half of baseline approaches have mean return close to 0.0 on these tasks), we conduct experiments mainly on FARMER, INF-DOORKEY, and INF-HARVESTER. The testing environments will have expected horizon lengths that are 2, 4, 8, and 16 times longer than those of the training environments. Additionally, the rewards for picking or placing markers, as well as the penalties for actions, will be scaled down by factors of 2, 4, 8, and 16, respectively, ensuring that the maximum accumulated reward for each task is normalized to 1. The detailed settings and experimental results for each of these three tasks are presented below.

### C.1   FARMER

During the training phase of all baselines, the maximum iteration number is set to 10. Afterward, we modified this number to 20, 40, 80, and 160 in the testing phase. As shown in Figure 9, as the expected horizon length increases, the performance of all baselines except HIPO declines significantly. This

indicates that our method has a much better inductive generalization property for this task. Further details on the definition of the maximum iteration number are provided in Section H.

## C.2 INF-HARVESTER

During the training phase of all baslines, the emerging probability is set to $\frac{1}{2}$. Following this, we modified this number to $\frac{3}{4}$, $\frac{7}{8}$, $\frac{15}{16}$ and $\frac{31}{32}$ in the testing phase. As shown in Figure 9, as the expected horizon length grows, the performances of HIPO, PAO, and DRL drop slightly, but the performances of LEAPS and HPRL drop extensively. Also, HC performs better when the scale increases, but drops when the scale grows further. More details on the definition of the emerging probability are provided in Section H.

## C.3 INF-DOORKEY

During the training phase of all baselines, the upper limit number of marker-picking and marker-placing is set to 16. Then, we modified this number to 32, 64, 128, and 256 in the testing phase. As shown in Figure 9, as the expected horizon length grows, the performance of all the baselines drop significantly. Nevertheless, HIPO has a minor performance drop compared to other baselines. Further details on the definition of the upper limit number are provided in Section H.

## D State machine extraction

In our approach, since the high-level neural network policy is incorporated, the proposed HIPO is only partially or locally interpretable – once the the high-level policy selects a programmatic option, human users can recognize the following execution of the program.

To further increase the interpretability of the trained high-level policy $f_\phi$, we extracted a state machine structure by the approach proposed in [36]. In this setup, since HIPO utilizes the previous programmatic option as input and predicts the following option, we focus only on encoding the environment state observations. Each state observation is processed by convolutional neural networks and fully connected layers into a $1 \times 32$ vector, which is then quantized into a $1 \times h$ vector. The hyperparameter $h$ balances between the simplicity of the finite state machine and performance drop. Using these quantized vectors, we can construct a state-transition table. The final step involves minimizing these quantized vectors to effectively represent the state machine's structure. Examples of the extracted state machines are shown in Figure 10, Figure 11, and Figure 12.

## E Hyperparameters and experimental settings

### E.1 HIPO

#### E.1.1 Encoder & decoder

We follow the training procedure and the model structure proposed in [74], which uses the GRU [15] network to implement both the encoder $q_\psi$ and the decoder $p_\theta$ with hidden dimensions of 256. The encoder $q_\psi$ and decoder $p_\theta$ are trained on programs randomly sampled from the Karel DSL. The

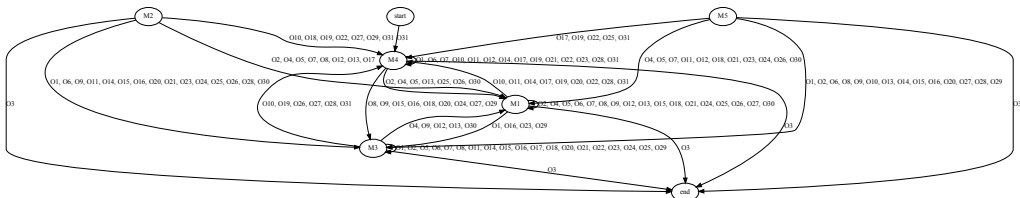

Figure 10: **Example of extracted state machine on FARMER**. $O1$ to $O31$ represent the unique quantized vectors encoded from observations. The corresponding programs of $M1$ to $M5$ are displayed in Figure 23.

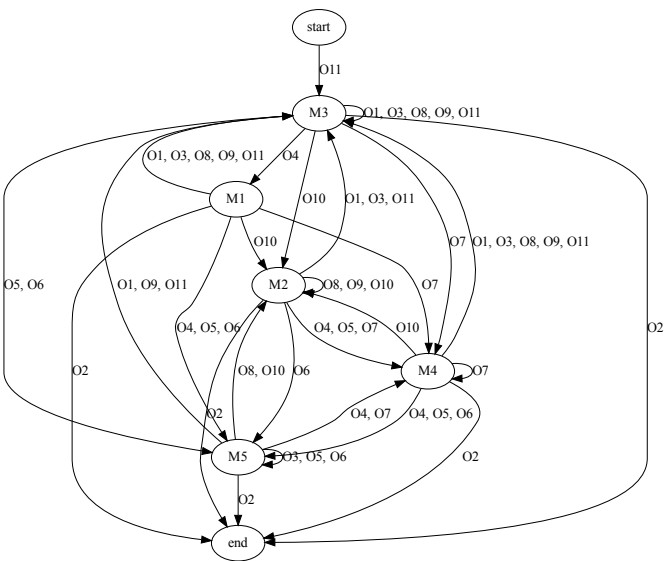

Figure 11: **Example of extracted state machine on INF-DOORKEY**. $O1$ to $O11$ represent the unique quantized vectors encoded from observations. The corresponding programs of $M1$ to $M5$ are displayed in Figure 24.

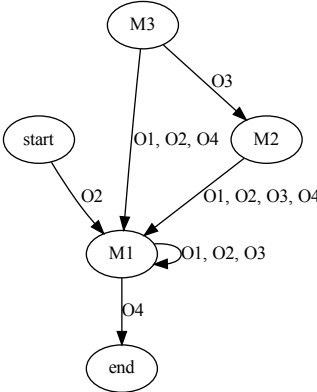

Figure 12: **Example of extracted state machine on INF-HARVESTER**. $O1$ to $O4$ represent the unique quantized vectors encoded from observations. The corresponding programs of $M1$ to $M3$ are displayed in Figure 25.

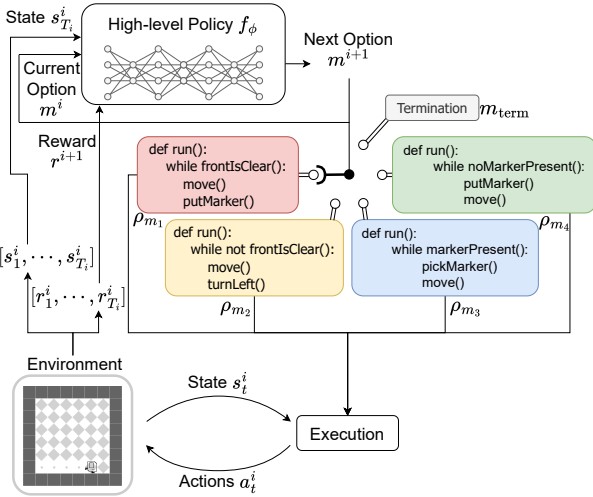

Figure 13: **High-level policy execution.**

loss function for training the encoder-decoder model integrates the $\beta$-VAE [27] loss, the program behavior reconstruction loss [74], and the latent behavior reconstruction loss [74].

The program dataset used to train $q_\psi$ and $p_\theta$ consists of 35,000 programs for training and 7,500 programs for validation and testing. Program tokens are sequentially sampled for each program based on defined probabilities until an ending token is reached or the maximum program length of 40 is attained. Defined probabilities of each type of token are listed below:

- `WHILE`: 0.15
- `REPEAT`: 0.03
- `STMT_STMT`: 0.5
- `ACTION`: 0.2
- `IF`: 0.08
- `IFELSE`: 0.04

Note: the token `STMT_STMT` represents a division operation that splits the current token into two separate tokens, which are then sampled following the above probability list. This token primarily dictates the program's length, as well as the quantity and complexity of nested loops and statements.

### E.1.2 Programmatic options

We conduct the procedure described in Section A.3.3 to search programmatic options for each high-level policy. For tasks only involving skills of marker-picking and traversal, i.e., INF-HARVESTER, SEESAW, and UP-N-DOWN, we designate the number of programmatic options as $|M| = 3$, while for the remaining tasks, $|M| = 5$.

### E.1.3 High-level policy

The high-level policy $f_\phi$, whose training process is illustrated in Figure 13, comprises convolutional layers [23, 37] to extract features from the Karel states and fully connected layers to determine the next program for execution. Meanwhile, one-hot encodings are used by the high-level policy to represent the indices of programmatic options. The detailed setting of the convolutional layers accords with the description in Section E.3. PPO [61] algorithm is adopted to optimize the high-level policy $f_\phi$. Hyperparameters are listed below:

- Maximum program number: 1000
- Batch size : 32
- Clipping: 0.05

- $\alpha$: 0.99
- $\gamma$: 0.99
- GAE lambda: 0.95
- Value function coefficient: 0.5
- Entropy coefficient: 0.1
- Number of updates per training iteration: 4
- Number of environment steps per set of training iterations: 32
- Number of parallel actors: 32
- Optimizer : Adam
- Learning rate: {0.1, 0.01, 0.001, 0.0001, 0.00001}

## E.2 PAO

The implemented PAO resembles the setting in Section E.1. The input, output, and structure of the high-level policy $f_\phi$ remain the same. However, the 5 low-level programs are replaced by 5 primitive actions (`move`, `turnLeft`, `turnRight`, `putMarker`, `pickMarker`).

## E.3 DRL

DRL training on the Karel environment is implemented with the PPO [61] algorithm with 20 million timesteps. Both the policy and value networks share a convolutional encoder interpreting the state of the grid world. This encoder comprises two layers. The first layer comprises 32-filters, a size-4 kernel, and a stride of 1. The following layer has 32 filters, a size-2 kernel, and the same stride of 1. The high-level policy neural network yields the probability distribution among primitive actions (move, turnLeft, turnRight, putMarker, pickMarker) and termination, given the current environmental state as input. In our experiments with DRL on Karel-Long tasks, all hyperparameters are fixed as listed below, except for learning rates tuned in a grid-search manner.

- Maximum horizon: 50000
- Batch size : 32
- Clipping: 0.05
- $\alpha$: 0.99
- $\gamma$: 0.99
- GAE lambda: 0.95
- Value function coefficient: 0.5
- Entropy coefficient: 0.1
- Number of updates per training iteration: 4
- Number of environment steps per set of training iterations: 128
- Number of parallel actors: 32
- Optimizer : Adam
- Learning rate: {0.1, 0.01, 0.001, 0.0001, 0.00001}

## E.4 Option-Critic

Option-Critic training on the Karel environment is implemented with the option-critic architecture [4]. algorithm with 20 million timesteps. The raw state inputs are extracted by an encoder comprised of two layers. The first layer comprises 32 filters, a size-4 kernel, and a stride of 1. The following layer has 32 filters, a size-2 kernel, and the same stride of 1. In our experiments with Option-Critic on Karel-Long tasks, all hyperparameters are listed below.

- Maximum horizon: {2000, 4000, 6000, 8000, 10000}
- Number of options: {2, 3, 4, 5}

- Batch size : {32, 64, 128}
- $\gamma$: 0.99
- Termination Regularization: {0.1, 0.01}
- Freeze interval: {128, 256, 512, 1024}
- Update frequency: {32, 64, 128}
- Entropy regularization: {0.1, 0.01}
- Learning rate: {0.1, 0.01, 0.001, 0.0001, 0.00001}

### E.5 LEAPS

Following the setup detailed in [74], we conducted experiments with various CEM-related hyperparameters as listed below, in order to optimize rewards for LEAPS.

- Population size (n): {8, 16, 32, 64}
- $\sigma$: {0.1, 0.25, 0.5}
- $e$: {0.05, 0.1, 0.2}
- Exponential $\sigma$ decay: {True, False}
- Initial distribution $\mathcal{P}$ : $\{\mathcal{N}(1,0), \mathcal{N}(0,\sigma), \mathcal{N}(0,0.1\sigma)\}$

### E.6 HPRL

Aligned with the approach described in [46], we trained the meta-policy for each task to predict a program sequence, with hyperparameters listed below.

- Max subprogram: 5
- Max subprogram Length: 40
- Batch size : 128
- Clipping: 0.05
- $\alpha$: 0.99
- $\gamma$: 0.99
- GAE lambda: 0.95
- Value function coefficient: 0.5
- Entropy coefficient: 0.1
- Number of updates per training iteration: 4
- Number of environment steps per set of training iterations: 32
- Number of parallel actors: 32
- Optimizer : Adam
- Learning rate: 0.00001
- Training steps: 25M

### E.7 HC

According to [10], we performed the stochastic hill climbing search to find the resulting program. Hyperparameters are listed below.

- Number of neighbor candidate programs: 250
- Number of parallel actors: 16
- Total number of evaluated programs: 1000000
- Maximum function calls: 10000

# F  Details of KAREL problem set

Introduced in Trivedi et al. [74], the KAREL problem set includes 6 tasks: STAIRCLIMBER, FOUR-CORNER, TOPOFF, MAZE, CLEANHOUSE and HARVESTER. Figure 14 and Figure 15 depict randomly generated initial states, legit sampled internal states, and desired final states for each task. The experimental results in Table 1 average 32 rewards obtained across 32 randomly generated initial states of the environment.

## F.1  STAIRCLIMBER

In a $12 \times 12$ grid environment, the agent's task is to climb the stairs and reach designated marked grid. Both the agent's starting position and the marked grid's position are randomly initialized on the stairs, with the marked grid always placed at the higher end. The reward is sparsely defined as: the agent receives a reward of 1 for reaching the goal, $-1$ for moving off the stairs, and 0 otherwise.

## F.2  FOURCORNER

In a $12 \times 12$ grid environment, the agent is tasked with placing a marker at each of the four corners. The agent receives no reward for placing a marker anywhere on the grid other than the four corners. Each successful marker placement at an unmarked corner earns a reward of $0.25$.

## F.3  TOPOFF

In a $12 \times 12$ grid environment, the agent's objective is to place a marker on every grid cell in columns where the bottom row has a marker in presence. Moreover, the agent should end up in the rightmost square of the row at the end of the episode. The agent receives rewards for consecutively and correctly placing markers until a mistake, which occurs if it places a marker on an empty cell in the bottom row or a marked cell.

## F.4  MAZE

In an $8 \times 8$ grid environment, the agent is tasked to find a marker by navigating the grid environment. The marker's location, the initial agent position, and the maze layout configuration are randomly initialized. The agent receives a sparse reward of 1 for successfully finding the marker in the environment, and 0 otherwise.

## F.5  CLEANHOUSE

In a $14 \times 22$ grid environment, the agent's target is to collect scattered markers as many as possible. The initial location of the agent is fixed, whereas the scattered markers are randomly placed within the environment, adjacent to walls. The return is calculated as the ratio of the number of collected markers to the total number of initially designated markers.

## F.6  HARVESTER

In an $8 \times 8$ grid environment, initially populated with markers in all grid cells, the agent's objective is to pick up a marker from each location within. The return is calculated as ratio of the number of picked markers to the total number of initially placed markers.

# G  Details of KAREL-HARD problem set

The KAREL-HARD problem set proposed by Liu et al. [46] involves 5 tasks: DOORKEY, ONE-STROKE, SEEDER and SNAKE. Each task in this benchmark is designed to be more constrained and structurally complex than those in the KAREL problem set. Figure 21 depicts randomly generated initial states, legit sampled internal states, and final states for each task. The experimental results in Table 1 average 32 rewards obtained across 32 randomly generated initial states of the environment.

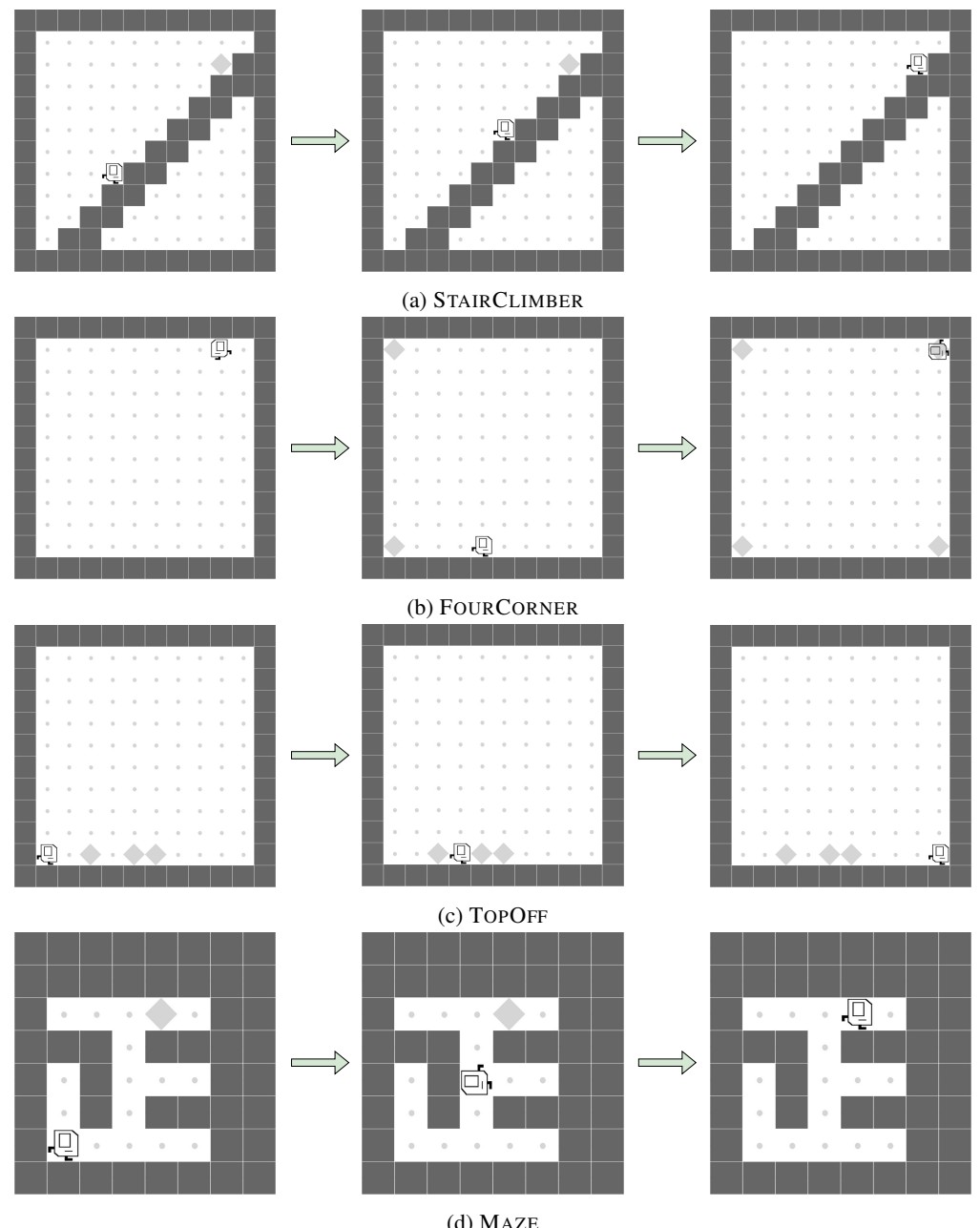

(a) STAIRCLIMBER

(b) FOURCORNER

(c) TOPOFF

(d) MAZE

Figure 14: Visualization of STAIRCLIMBER, FOURCORNER, TOPOFF, and MAZE in the KAREL problem set presented in Trivedi et al. [74]. For each task, a random initial state, a legitimate internal state, and the ideal end state are shown. In most tasks, the position of markers and the initial location of the Karel agent are randomized. More details of the KAREL problem set can be found in Section F.

## G.1 DOORKEY

In an $8 \times 8$ grid environment, partitioned into a $6 \times 3$ left room and a $6 \times 2$ right room by a column of walls, the agent's objective is to collect a key (represented by a marker) in the left room to unlock a door (an empty grid cell subsequently occurs in that wall column) and place the key on a marked grid cell in the right room. The agent's initial location, the key's location, and the target's location are randomly initialized. The agent receives a reward of $0.5$ for collecting the key and another $0.5$ for placing the key on the target.

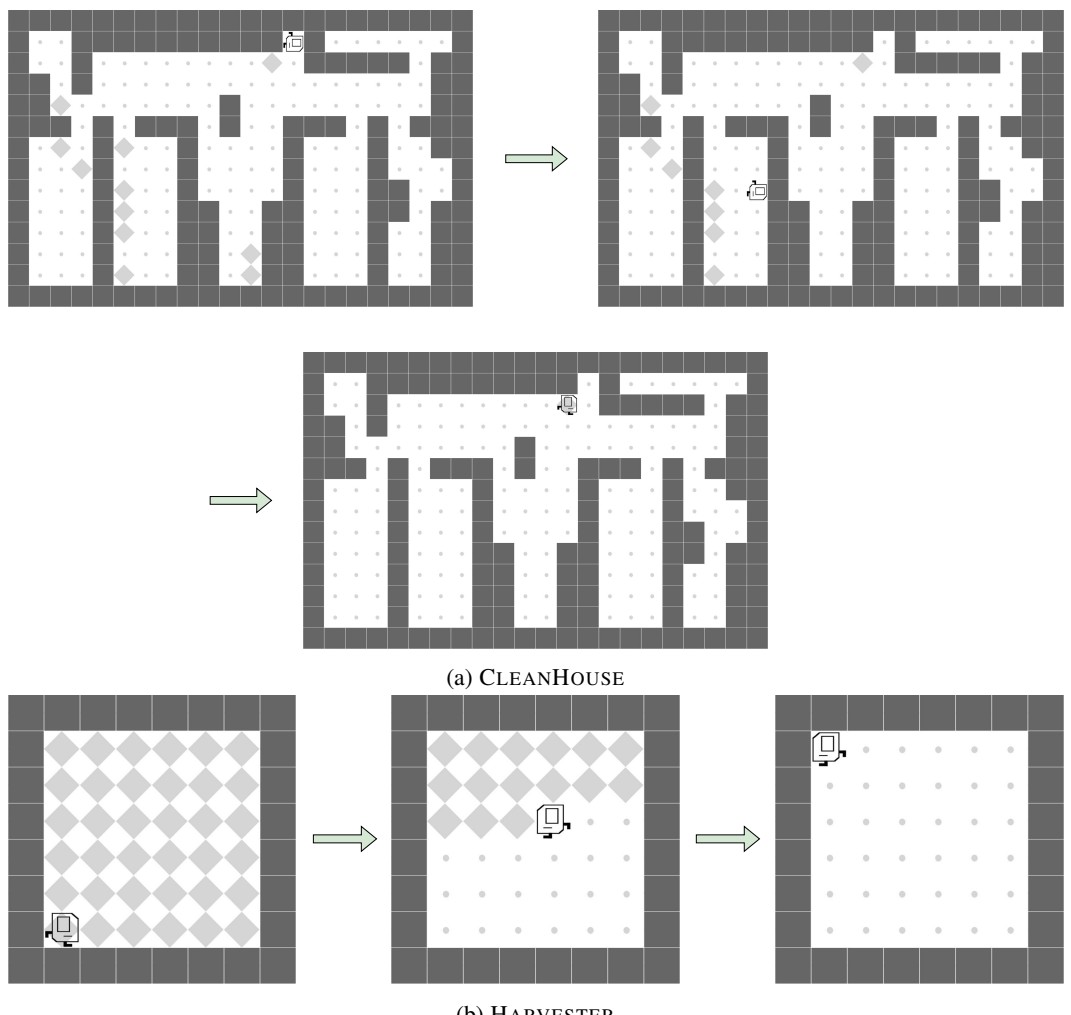

(a) CLEANHOUSE

(b) HARVESTER

Figure 15: Visualization of CLEANHOUSE and HARVESTER in the KAREL problem set presented in Trivedi et al. [74]. For each task, a random initial state, a legitimate internal state, and the ideal end state are shown. More details of the KAREL problem set can be found in Section F.

## G.2 ONESTROKE

In an $8 \times 8$ grid environment, the agent is tasked to navigate through all grid cells without revisiting any of them. An empty grid cell is replaced with a wall once visited. The episode ends once the agent collides with any of these walls. The return is calculated as the ratio of visited grids to the total number of empty grids in the initial environment.

## G.3 SEEDER

In an $8 \times 8$ grid environment, the agent's task is to place one marker on every grid cell. The episode terminates immediately whenever the agent places multiple markers on one grid cell. The return is calculated as the ratio of the number of successfully placed markers to the total number of empty grids in the initial environment.

## G.4 SNAKE

In an $8 \times 8$ grid environment, the agent operates as the snake's head and seeks to consume (pass-through) as much food (markers) as possible without colliding with its own body. The snake's body grows by 1 grid cell for each marker consumption, and a new marker appears at a different location.

Throughout the episode, only one marker exists in the environment. The reward is defined as $\frac{1}{20}$, which is the reciprocal of the targeted number of marker consumption, for each marker consumption.

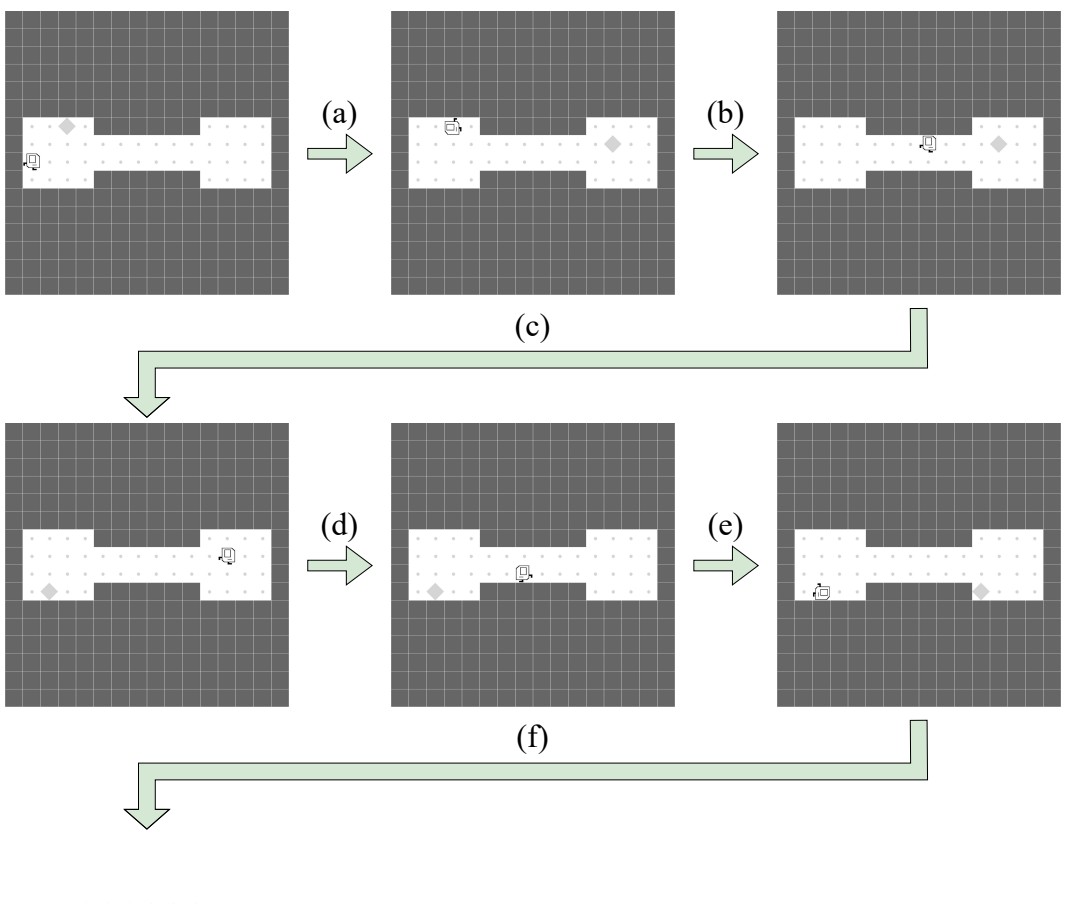

Figure 16: **Visualization of SEESAW in the KAREL-LONG problem set.** This figure partially illustrates a typical trajectory of the Karel agent during the task SEESAW. (a): Once the Karel agent collects a marker in the left chamber, a new marker appears in the right chamber. (b): The agent must navigate through the central corridor to collect the marker in the right chamber. (c): Once the Karel agent collects a marker in the right chamber, a new marker further appears in the left chamber. (d): Once again, the agent is traversing through the corridor to the left chamber. (e): A new marker appears in the right chamber again after the agent picks up the marker in the left chamber. (f): The agent will move back and forth between the two chambers to collect the emerging markers continuously. Note that the locations of all the emerging markers are randomized. Also, note that we have set the number of emerging markers to 64 during the training phase (*i.e.*, the agent has to pick up 64 markers to fully complete the task.) More details of the task SEESAW can be found in Section H.

## H    Details of KAREL-LONG problem set

We introduce the novel KAREL-LONG problem set as a benchmark to evaluate the capability of HIPO. Each task in our KAREL-LONG benchmark is crafted to exhibit long-horizon characteristics derived from Karel states. Additionally, we ensure that these tasks maintain a constant per-action cost, set at 0.0001. Figure 16, Figure 17, Figure 18, Figure 19, and Figure 20 depict the tasks within the KAREL-LONG problem set, exhibiting randomly generated initial states and several internal states sampled from legitimate trajectories.

### H.1 SEESAW

In a $16 \times 16$ grid environment, the agent's objective is to traverse back and forth between two $4 \times 4$ chambers (the left chamber and the right chamber), continuously collecting markers. To facilitate movement between the left and right chambers, the agent must traverse through a central $2 \times 6$ corridor. Initially, one marker is positioned in the left chamber, awaiting collection by the agent. Upon picking a marker in a chamber, another is randomly generated in the opposite chamber, continuing the agent's task of collecting markers. Consequently, the agent must navigate between the two chambers to collect markers continuously. The return is defined as the ratio of the number of picked markers to the total number of markers generated by the environment, termed as "emerging markers."

### H.2 UP-N-DOWN

In an $8 \times 8$ grid, the agent's objective is to ascend and dscend the stairs repeatedly to collect markers (loads). Once a marker below/above the stairs is picked up, the next marker will appear above/below the stairs, and so on. The agent would receive an additional constant penalty (i.e., $-0.005$) for being out of contact with the stair. The return is defined as the ratio of the number of picked markers to the total number of markers generated by the environment, termed as "emerging loads."

### H.3 FARMER

In an $8 \times 8$ grid environment, the agent's objective is twofold: initially, to fill the entire layout with markers, and subsequently, to collect them. In the initial state, all girds, except for the one in the upper-right corner, are empty, signaling the agent to proceed to populating the layout with markers, akin to farmer sowing seeds. After most grids are marked, the agent deploys the harvesting phase, analogous to a farmer collecting crops. Following this, the agent is prompted to refill the environment again, and the cycle repeats. We've imposed a maximum iteration limit to represent the anticipated number of filling-and-collecting rounds. The return is calculated as the ratio of the number of picked-and-placed markers to the total theoretical capacity of picked-and-placed markers, termed as "max markers."

### H.4 INF-DOORKEY

In an $8 \times 8$ grid environment, partitioned into 4 chambers, the agent is tasked to pick up markers in certain chambers, place markers in others, and continuously traverse between chambers until a predetermined upper limit number of marker-picking and marker-placing is reached. As mentioned earlier, the environment is partitioned into 4 chambers, restricting the agent's placement (or pick-up) actions to one chamber at a time. Upon completing a placement (or pick-up) action in a chamber, the passage to the next chamber opens (an emptied grid cell), allowing the agent to advance and perform another action. The return is calculated as the ratio of the number of picked-and-placed markers to the total number of markers that the agent can theoretically pick and place, referred to as "max keys."

### H.5 INF-HARVESTER

In a $16 \times 16$ grid environment, the agent's goal is to continuously pick up markers until none remain and no further markers emerge. Initially, the environment is fully stocked with markers. Each time the agent picks up a marker, there is a probability, termed the "emerging probability," that a new marker will appear in an empty grid cell, granting the agent to collect markers continuously and indefinitely. The return is calculated as the ratio of the number of picked markers to the expected total number of markers the environment can generate given a specific emerging probability.

## I  Designing domain-specific languages

Our program policies are designed to describe high-level task-solving procedures or decision-making logics of an agent. Therefore, our principle of designing domain-specific languages (DSLs) considers a general setting where an agent can perceive and interact with the environment to solve specific tasks. DSLs integrate control flows, perceptions, and actions. While control flows are domain-independent, perceptions and actions can be designed based on the domain of interest, requiring specific expertise and domain knowledge.

Table 3: **Performance of programmatic policies on KAREL-LONG across thirty-two random seeds.** Mean return and standard deviation of all methods across the KAREL-LONG problem set, evaluated over thirty-two random seeds.

| Method | SEESAW | UP-N-DOWN | FARMER | INF-DOORKEY | INF-HARVESTER |
|--------|--------|-----------|--------|-------------|---------------|
| LEAPS | $0.00 \pm 0.01$ | $0.01 \pm 0.01$ | $0.02 \pm 0.00$ | $0.00 \pm 0.01$ | $0.11 \pm 0.00$ |
| HPRL | $0.00 \pm 0.00$ | $0.00 \pm 0.00$ | $0.02 \pm 0.00$ | $0.01 \pm 0.01$ | $0.49 \pm 0.07$ |
| HC | $0.23 \pm 0.10$ | $0.14 \pm 0.27$ | $0.24 \pm 0.06$ | $0.21 \pm 0.13$ | $\mathbf{0.89} \pm 0.00$ |
| HIPO (Ours) | $\mathbf{0.50} \pm 0.14$ | $\mathbf{0.72} \pm 0.07$ | $\mathbf{0.51} \pm 0.22$ | $\mathbf{0.65} \pm 0.08$ | $0.76 \pm 0.04$ |

Such DSLs are proposed and utilized in various domains, including ViZDoom [34], 2D MineCraft [3, 71], and gym-minigrid [14]. Recent works [43, 79] also explored describing agents' behaviors using programs with function-taking arguments.

# J    Limitations

While the Hierarchical Programmatic Option (HIPO) Framework provides notable advantages including interpretability, long-horizon task solving, inductive generalizability, and sample efficiency, it also has limitations that merit consideration in comparison to other methods. Techniques utilizing first-order logic programming [19, 33, 50, 83], LLM-based interpretation [49], and decision tree policies [7, 35] are also specifically designed to create interpretable, explainable, and generalizable agents for solving MDPs.

In environments with a large state or action spaces, neuro-symbolic methods adopting symbolic policies [19, 33, 50, 83] are generally less computationally intensive and more feasible for training compared to programmatic approaches like this work and [10, 46, 74]. However, symbolic policies [19, 33, 50, 83] are inherently less intepretable, whereas programmatic policies, which could convey the logical flow of reasoning, are more intuitive to general users.

While these symbolic policies are often complex for the average user, INSIGHT [49] uses LLMs to explain symbolic policies (i.e. polynomials with trainable parameters), thereby enhancing interpretability. This is something we could consider for future improvements, like using LLMs to explain programmatic options for better interpretability. Nonetheless, LLM prompting requires detailed environment and task descriptions, where the generated explanation is dependent on the LLM's understanding of the environment and the quality of the descriptions. In contrast, HIPO directly derives an interpretable programmatic policy through interaction with the environment, avoiding the aforementioned issue.

Other proposals for enhancing interpretability involve decision trees. Both [7, 35] focus on learning decision trees within the imitation learning framework, which includes a DNN oracle for expert demonstration. In [35], learned decision trees are further converted into if-else Python programs. However, the interpretability and performance of these methods depend heavily on the tree learning algorithm. Overly complex or brute-force programs become less interpretable and meaningful. Moreover, these methods generally require specific state information (e.g. object coordinates) for predetermined training, making it challenging to identify the necessary symbolic features for optimal performance or to develop a plausible training scheme. In contrast, LEAPS and its successors search only within the embedding space and perceive the state space partially and consistently, operating with fewer assumptions and achieving good performance.

Besides, in our experiments detailed in Section 5, HIPO outperforms other baselines, including DRL, where deep neural networks were used as expert policies in [7, 35] to demonstrate trajectories for learning decision trees. This suggests that decision-tree methods [7, 35] are unable to surpass our HIPO framework due to the limitations of the underperforming DNN expert. In [7], the authors propose methods for verifying decision-tree policies. The structure of decision trees contributes to efficient verification, allowing for quantitative assessment of the policy's correctness, stability, and robustness. This process aims to evaluate the effectiveness of the policy, including its performance in various scenarios and the presence of other desirable properties. Developing a method to quantitatively verify a programmatic policy remains a significant challenge.

Finally, while HIPO retrieves a set of effective and compatible programs through a search method, exploring approaches that leverage prior knowledge, *e.g.*, large language models [47] or offline datasets in imitation learning [12, 28, 30, 38, 41, 60] or skill-based reinforcement learning [40, 54, 57], to efficiently retrieve programs is a promising research direction.

## K Computational resources

For our experiments, we utilized the following workstation: 20-core Intel(R) Xeon(R) W-2255 CPU @ 3.70GHz, with 2X NVIDIA GeForce RTX 4070 Ti GPU

The rough computation time for the main experiments of our framework and baselines are as follows:

1. SEESAW
   (a) DRL: 24 hours
   (b) LEAPS: 48 hours
   (c) HPRL: 48 hours
   (d) HIPO: 48 hours

2. UP-N-DOWN:
   (a) DRL: 24 hours
   (b) LEAPS: 48 hours
   (c) HPRL: 48 hours
   (d) HIPO: 48 hours

3. FARMER:
   (a) DRL: 24 hours
   (b) LEAPS: 48 hours
   (c) HPRL: 48 hours
   (d) HIPO: 72 hours

4. INF-DOORKEY:
   (a) DRL: 24 hours
   (b) LEAPS: 48 hours
   (c) HPRL: 48 hours
   (d) HIPO: 72 hours

5. INF-HARVESTER:
   (a) DRL: 24 hours
   (b) LEAPS: 48 hours
   (c) HPRL: 48 hours
   (d) HIPO: 48 hours

## L Impact statements

Our work introduces Hierarchical Programmatic Option framework (HIPO), a novel framework that combines programmatic RL and HRL, enabling autonomous agents to represent complex behaviors and address long-term tasks. HIPO demonstrates significant potential in automating laborious or risky tasks, enhancing the efficiency and safety of autonomous systems. While offering promising societal benefits, we acknowledge the importance of addressing ethical considerations, including potential biases inherited from training data or adversarial attacks. Ongoing research in the field of responsible AI and fairness in machine learning aims to mitigate these biases, and we acknowledge the importance of continuous efforts to ensure that AI systems, including HIPO, adhere to ethical standards and contribute positively to society.

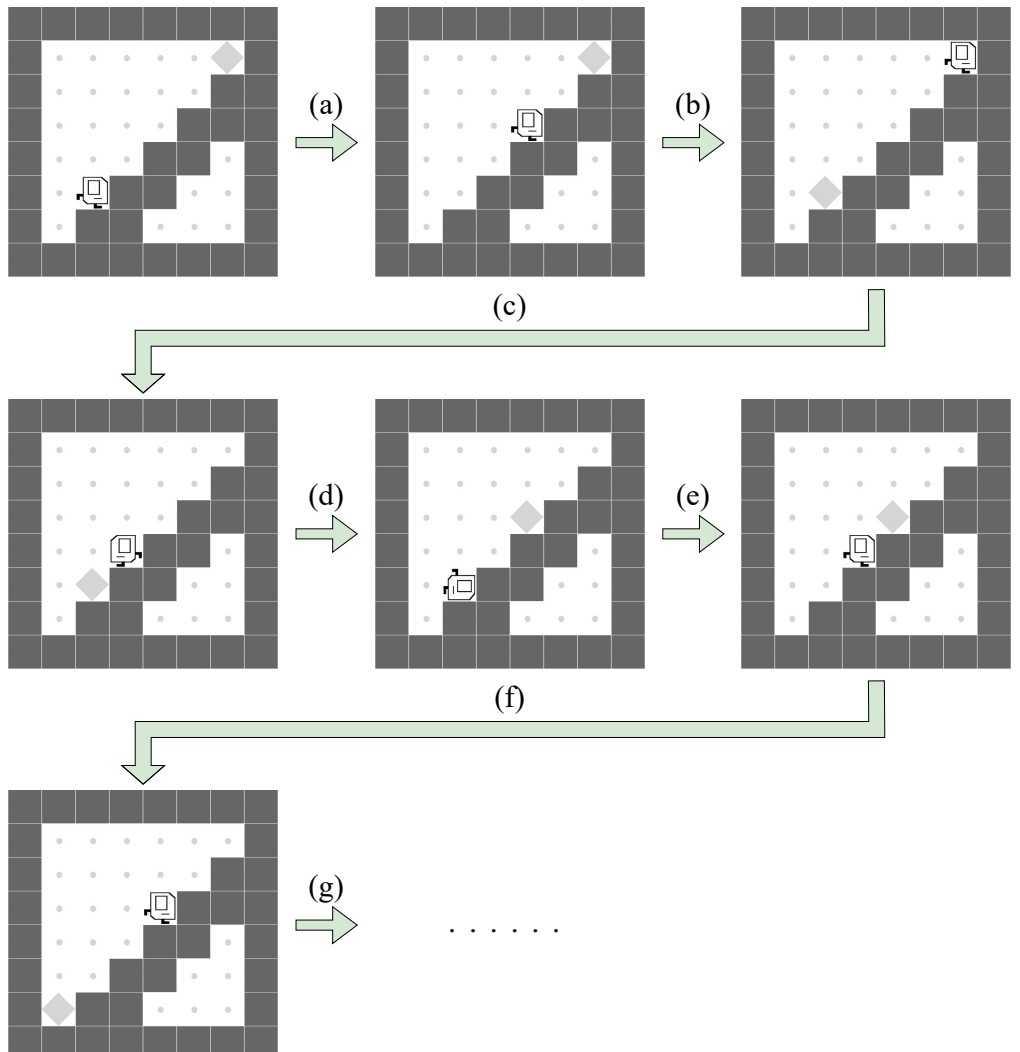

Figure 17: **Visualization of UP-N-DOWN in the KAREL-LONG problem set.** This figure partially illustrates a typical trajectory of the Karel agent during the task UP-N-DOWN. (a): The Karel agent is ascending the stairs to collect a load located above the stairs. Note that the agent can theoretically collect the load without directly climbing up the stairs, but it will receive some penalties for doing so. (b): Once the agent collects the load, a new load appears below the stairs. (c): The agent then descends the stairs to collect a load located below. Note that the agent can theoretically collect the load without directly climbing down the stairs, but it will receive some penalties for doing so. (d): Upon the agent collecting the load, a new load appears above the stairs. (e): The agent once again ascends the stairs to collect a load. (f): A new load appears below the stairs again after the agent collects the load located above. (g): The agent would continue to collect the emerging loads in descend-ascend cycles repeatedly on the stairs. Note that the locations of all the emerging loads are randomly initiated right next to the stairs. The load must appears below/above the stairs after the agent just finished ascending/descending. Also, we have fixed the number of emerging loads to 100 during the training phase (*i.e.*, the agent shall collect 100 loads to complete the task). More details of the task UP-N-DOWN can be found in Section H.

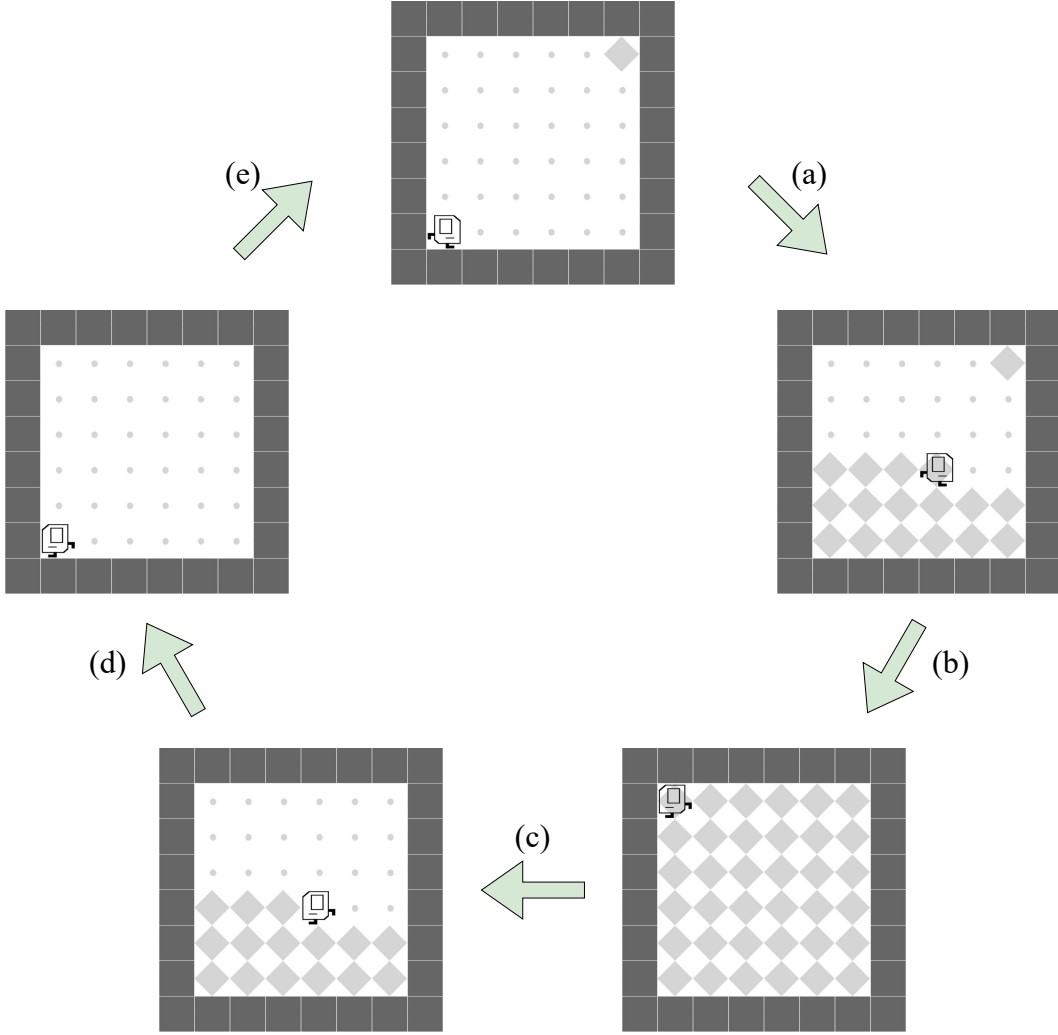

Figure 18: **Visualization of FARMER in the KAREL-LONG problem set.** This figure partially illustrates a typical trajectory of the Karel agent during the task FARMER. (a): The Karel agent is filling (placing) the entire environment layout with markers. In the initial state, there exists a single marker located in the upper-right corner. The marker is purposed to prompt the agent to start filling the environment layout. (b): The agent successfully populates the entire environment. (c): The agent is then asked to pick up markers as much as possible. (d): The agent successfully picks all markers up, leaving the environment empty. (e): If there is another filling-and-collecting round, a marker will appear in the upper-right corner to indicate that the agent should start the filling process again. Otherwise, the agent completes the entire task, and no further marker will appear. For simplicity, we only show the former case here. We have fixed the number of max markers to 720 during the training phase (*i.e.*, the agent has to fill the entire environment layout with markers and pick up all markers each 10 times to fully complete the task.) More details of the task FARMER can be found in Section H.

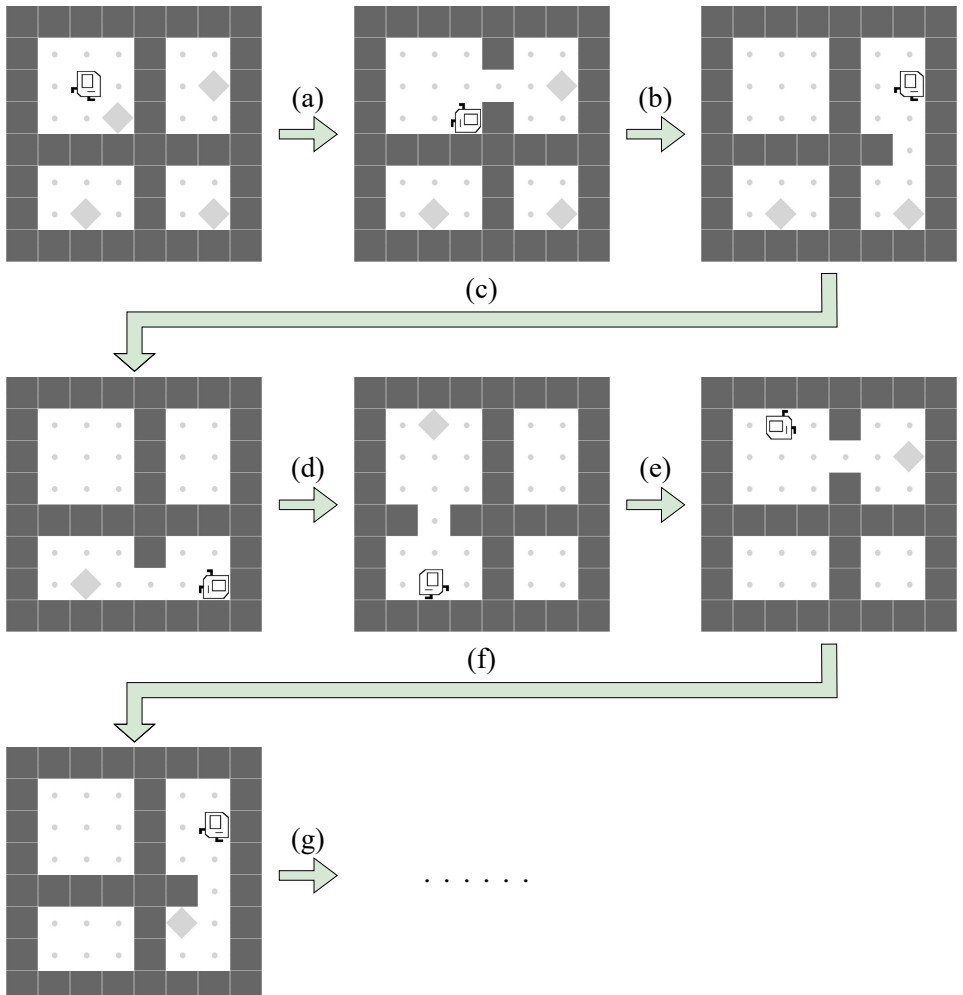

Figure 19: **Visualization of INF-DOORKEY in the KAREL-LONG problem set.** This figure partially illustrates a typical trajectory of the Karel agent during the task INF-DOORKEY. (a): The Karel agent picks up a marker in the upper-left chamber. Then, a passage to the upper-right chamber opens, allowing the agent to traverse through. (b): The agent successfully places a marker at a marked grid located in the upper-right chamber. Subsequently, a passage to the lower-right chamber opens, allowing the agent to traverse through. (c): After the agent collects a marker in the lower-right chamber, a passage to the lower-left chamber opens, allowing the agent to traverse through. (d): The agent properly places a marker at a marked grid located in the lower-left chamber. After that, a passage to the upper-left chamber opens, and a new marker appears in the upper-left chamber. (e): Upon the agent picking up a marker in the upper-left chamber, the passage to the upper-right chamber opens again, and a grid is marked randomly in the upper-right chamber. (f): The agent accurately places a marker at a marked grid located in the upper-right chamber. Afterward, the passage to the lower-right chamber opens again, and a new marker emerges in the lower-right chamber. (g): The agent will repeatedly pick up and place markers in this fashion until the number of max keys is reached. We have fixed the number of max keys to 100 during the training phase (*i.e.*, the agent has to pick and place 100 markers in total to fully complete the task.) More details of the task INF-DOORKEY can be found in Section H.

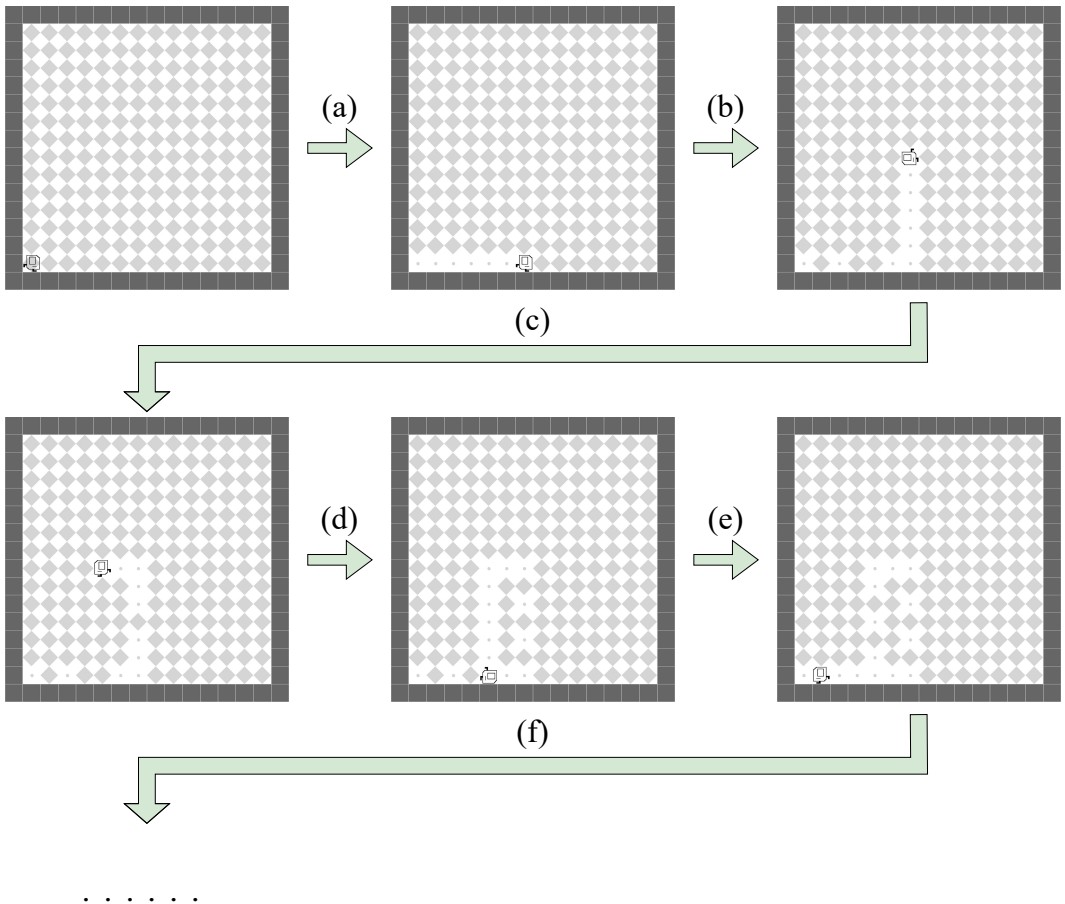

. . . . . .

Figure 20: **Visualization of INF-HARVESTER in the KAREL-LONG problem set.** This figure partially illustrates a legitimate trajectory of the Karel agent during the task INF-HARVESTER. (a): The Karel agent picks up markers in the last row. Meanwhile, no new markers are popped out in the last row. (b): The agent turns left and picks up 6 markers in the $7^{th}$ column while 3 markers appear in 3 previously empty grids in the last row. (c): The agent collects markers in the $8^{th}$ row while 1 marker appears in a previously empty grid in the $7^{th}$ column. (d): The agent picks up 6 markers in the $5^{th}$ column while 2 markers appear in 2 previously empty grids in the $7^{th}$ column. (e): The agent picks up 2 more markers in the last row while 2 markers appeared in 2 previously empty grids in the $5^{th}$ column. (f): Since markers appear in previously empty grids based on the emerging probability, the agent will continuously and indefinitely collect markers until none remain and no new markers appear in the environment. The emerging probability has been fixed to $\frac{1}{2}$ during the training phase. More details of the task INF-HARVESTER can be found in Section H.

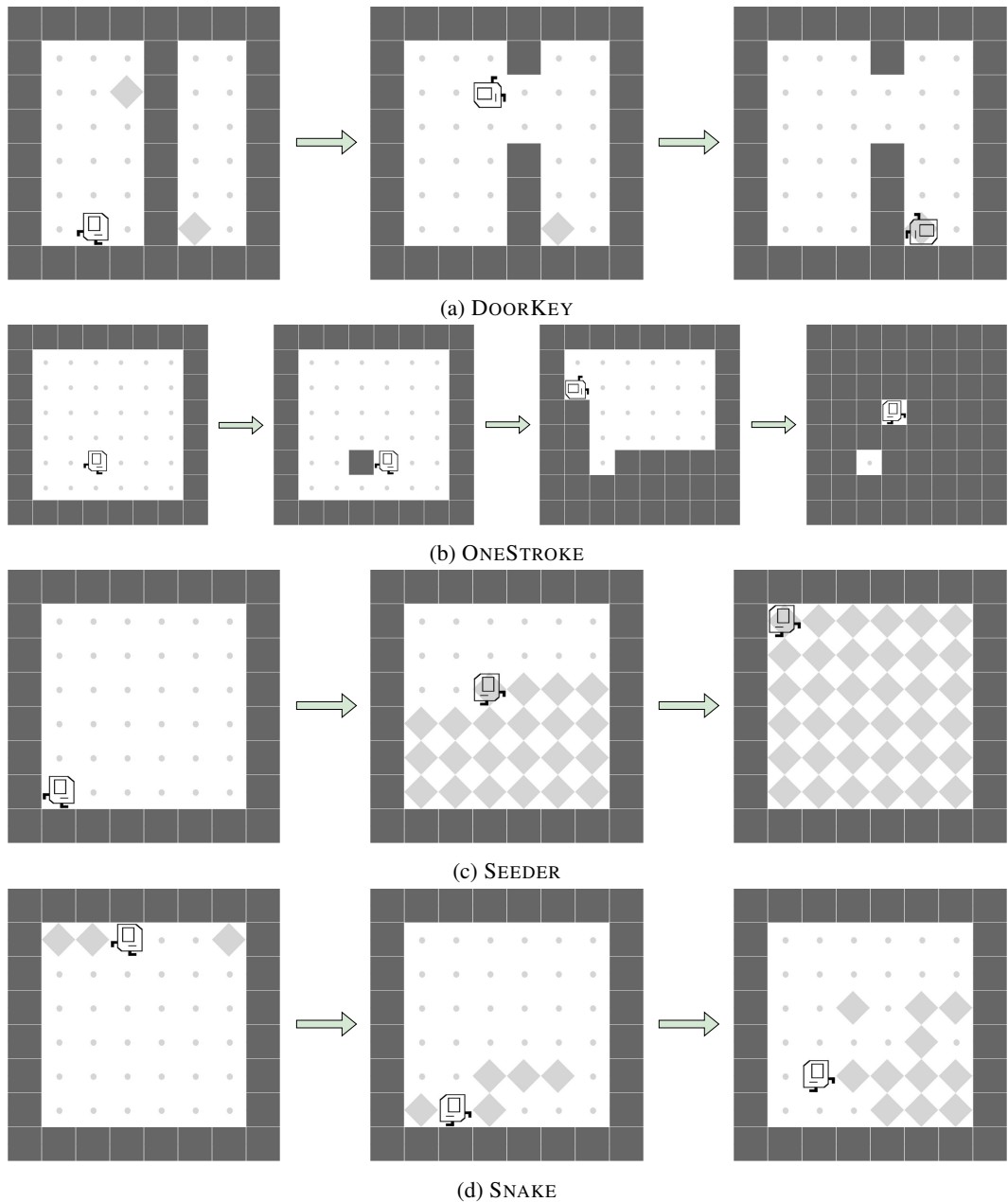

(a) DOORKEY

(b) ONESTROKE

(c) SEEDER

(d) SNAKE

Figure 21: Visualization of each task in the KAREL-HARD problem set proposed by Liu et al. [46]. For each task, a random initial state, some legitimate internal state(s), and the ideal end state are shown. More details of the KAREL-HARD problem set can be found in Section G.

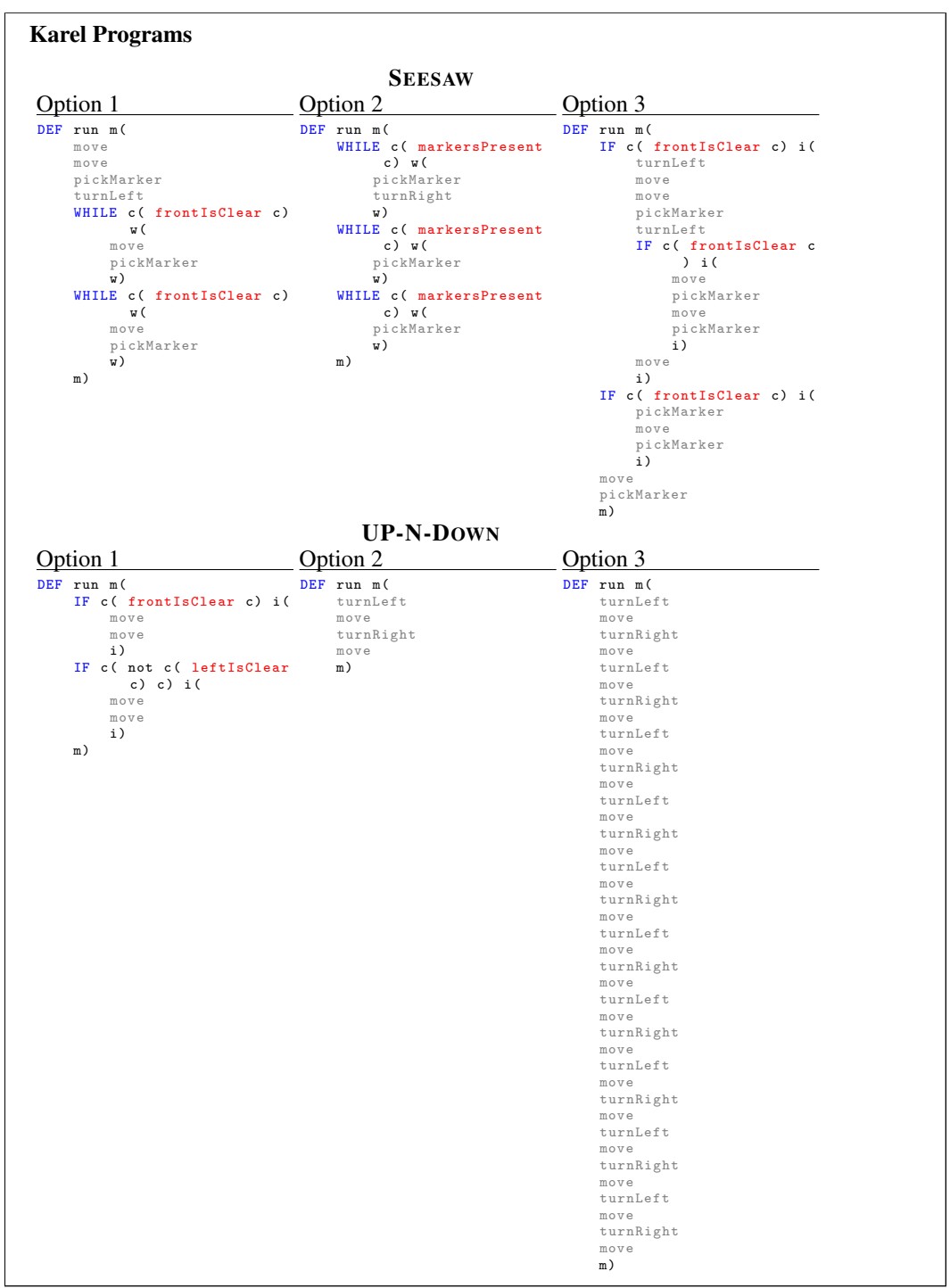

Figure 22: **Example programs on Karel-Long tasks: SEESAW and UP-N-DOWN.** The programs with best rewards out of five random seeds are shown. $|M| = 3$ for SEESAW and UP-N-DOWN.

**Karel Programs**

**FARMER**

Option 1

```
DEF run m(
    pickMarker
    REPEAT R=0 r(
        pickMarker
        move
        turnRight
        move
        move
        turnLeft
        move
        move
        turnRight
        move
        turnLeft
        move
        move
        pickMarker
        move
        turnLeft
        move
        move
        pickMarker
        move
        turnLeft
        move
        move
        pickMarker
        move
        r)
    m)
```

Option 2

```
DEF run m(
    turnRight
    move
    putMarker
    turnRight
    move
    pickMarker
    putMarker
    move
    pickMarker
    putMarker
    move
    pickMarker
    putMarker
    move
    pickMarker
    putMarker
    move
    pickMarker
    putMarker
    move
    m)
```

Option 3

```
DEF run m(
    turnRight
    move
    turnRight
    putMarker
    pickMarker
    move
    pickMarker
    putMarker
    move
    pickMarker
    putMarker
    move
    pickMarker
    putMarker
    move
    pickMarker
    putMarker
    move
    pickMarker
    putMarker
    move
    pickMarker
    m)
```

Option 4

```
DEF run m(
    REPEAT R=17 r(
        IF c( not c(
            rightIsClear c)
            c) i(
            putMarker
            move
            i)
        r)
    m)
```

Option 5

```
DEF run m(
    putMarker
    pickMarker
    turnRight
    move
    pickMarker
    turnRight
    move
    pickMarker
    putMarker
    move
    pickMarker
    putMarker
    move
    pickMarker
    putMarker
    move
    pickMarker
    putMarker
    move
    pickMarker
    putMarker
    move
    pickMarker
    putMarker
    move
    m)
```

Figure 23: **Example programs on Karel-Long tasks: FARMER.** The programs with best rewards out of five random seeds are shown. $|M| = 5$ for FARMER.

**Karel Programs**

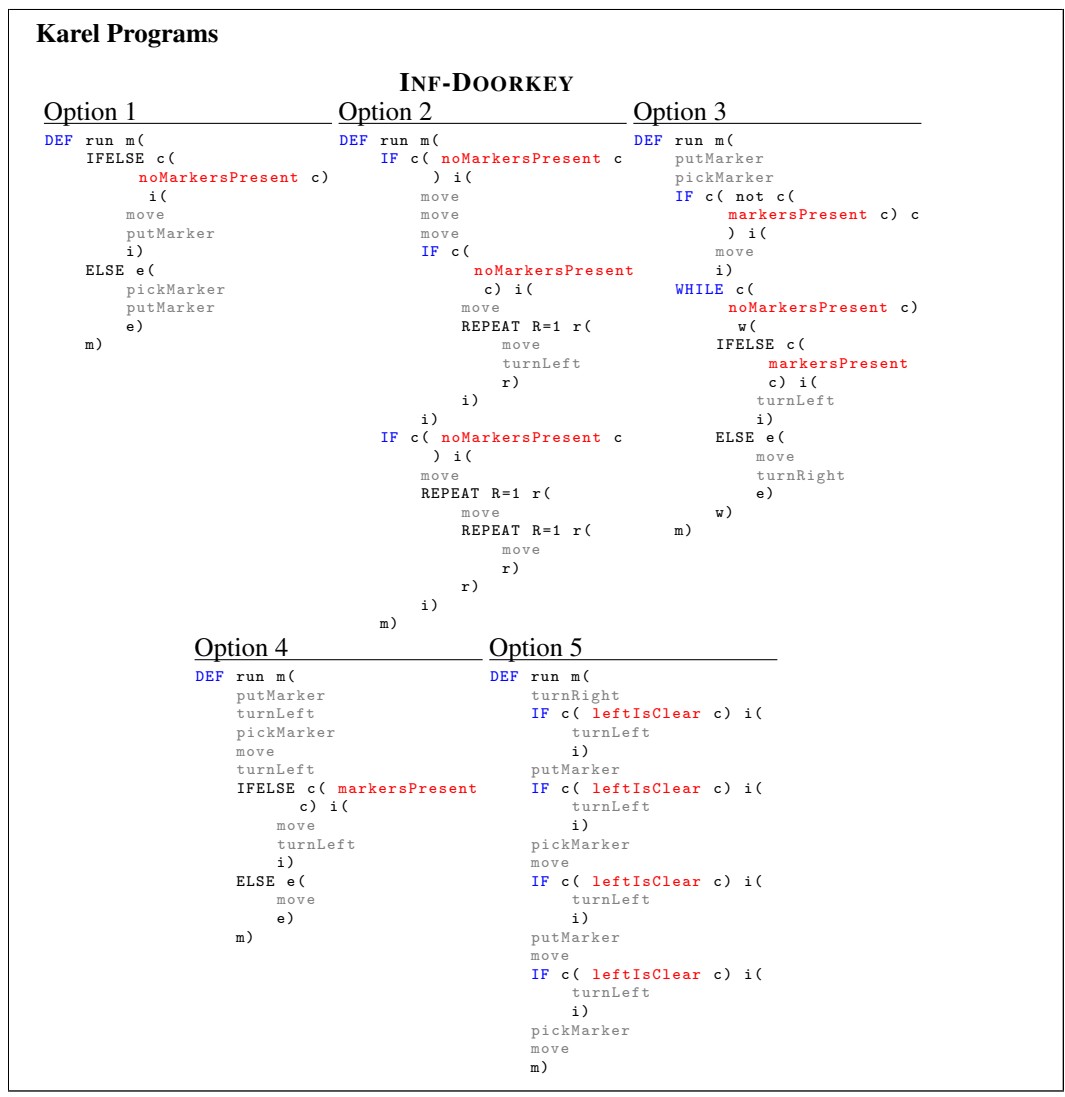

Figure 24: **Example programs on Karel-Long tasks: INF-DOORKEY.** The programs with best rewards out of five seeds are shown. $|M| = 5$ for INF-DOORKEY.

**Karel Programs**

**INF-HARVESTER**

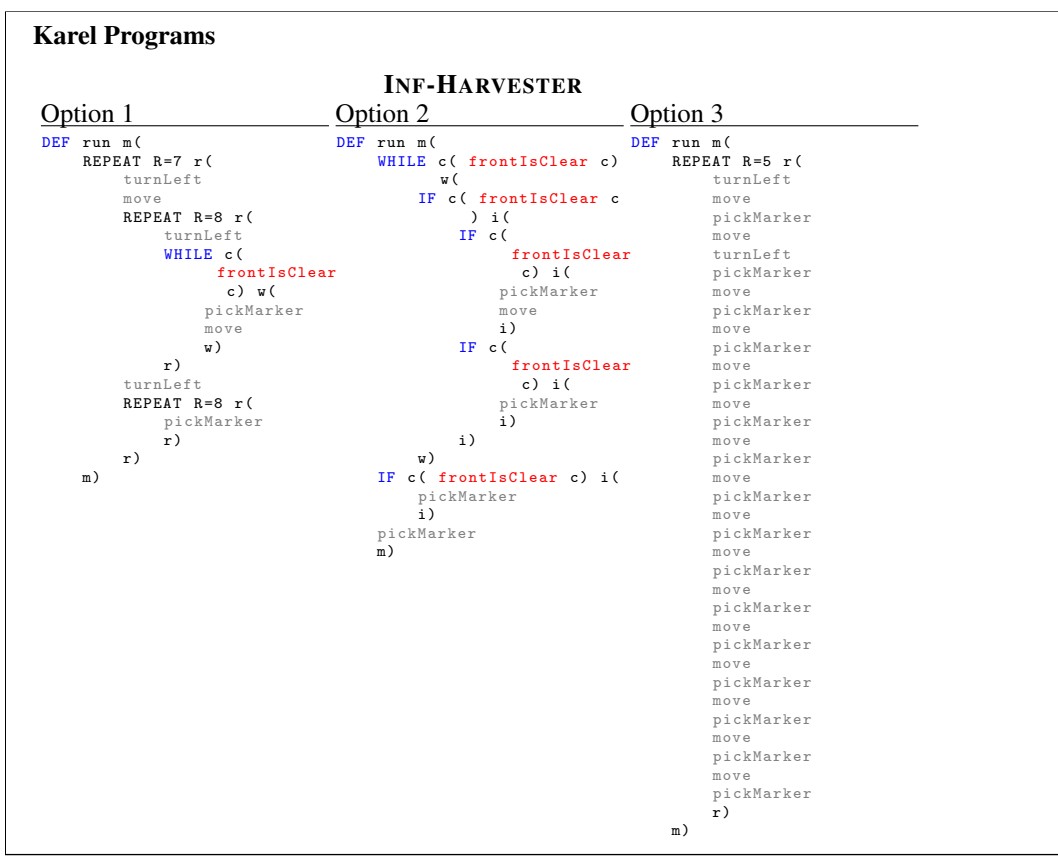

Figure 25: **Example programs on Karel-Long tasks: INF-HARVESTER.** The programs with best rewards out of five random seeds are shown. $|M| = 3$ for INF-HARVESTER.

