# OpenReview forum: "Hierarchical Programmatic Option Framework"
_NeurIPS.cc/2024/Conference — NeurIPS 2024 poster_

### Official Review · Reviewer_4qWN · 2024-07-04

**Soundness:** 4
**Presentation:** 4
**Contribution:** 3
**Rating:** 8
**Confidence:** 3

**Summary:**

This work builds off of deep reinforcement learning generating programmatic policies, and adapts it to solve long-horizon, repetitive tasks. Concretely, this work proposes HIPO, short for hierarchical programmatic option framework, which retrieves history programs by its neural embeddings, and uses these programs as reusable options to solve reoccurring tasks.

**Strengths:**

1) Effective approach: This work tackles an intuitively challenging problem and proposes a practically effective solution. Particularly on long-horizon tasks, as shown by the results in Karel-long in Table 2, demonstrates the effectiveness of the proposed method.
2) Thorough ablation studies: the paper conducts various ablation studies, and show the optimal performance of the proposed approach among other variants.

**Weaknesses:**

1) Lack of retrieval evaluation: The paper proposes special techniques to improve the effectiveness, diversity, and compatibility of retrieved programs, however, is there any empirical evidence to show that the designed modules do improve the effectiveness, diversity, and compatibility of programs?

**Questions:**

1) While generating reusable subroutines is intuitive, there are no examples on how these low-level policies look like throughout the paper, making it a bit harder to concretely visualize how the policies look like. Adding a few examples, even in appendix, would make the process more interpretable.
2) This work chooses a particular DSL (as introduced in Section 3), does the choice of DSL affect the performance of the proposed process? Would a less well-designed DSL cause the method to be ineffective?

**Limitations:**

Yes.

---

> ### Author Rebuttal · Authors · 2024-08-07
>
> We sincerely thank the reviewer for the thorough and constructive comments. Please find the response to your questions below.
>
> >Is there any empirical evidence to show that the designed modules do improve the effectiveness, diversity, and compatibility of programs?
>
> We thank the reviewer for this insight. We discuss the evaluation of effectiveness, diversity, and compatibility below.
>
> - Effectiveness: The ablation study in Section 5.3 and the experimental results presented in Table 1 serve as empirical evidences that demonstrate the efficacy of the proposed CEM+diversity searching paradigm.
> - Diversity: Section 5.7 (Figures 21, 22, 23, and 24) provides multiple examples of the retrieved option program sets that cover diverse skills and subroutines for each given task.
> - Compatibility: As suggested by the reviewer, to further investigate and quantify the compatibility among the retrieved programs, we conduct additional experiments of CEM+compatibility ×|M| (i.e., CEM with the evaluation function $G(z) = \frac{1}{D} \sum_{i=1}^{D} R_{\Psi_i}$, where the number of programs sequences $D$, the specified program sequence $\Psi_i$ and the normalized reward $R_{\Psi_i}$ are defined similarly as in Section 4.2.3 and Equation 1) for N = 1 time and select each of the result as the i-th option. Repeat the above process |M| times and take all |M| results as the set of programmatic options.
>
> | Method | Seesaw | UP-N-Down | Farmer | Inf-DoorKey | Inf-Harvester |
> |-|-|-|-|-|-|
> | CEM × \|M\| | 0.06 $\pm$ 0.10 | 0.39 $\pm$ 0.36 | 0.03 $\pm$ 0.00 | 0.11 $\pm$ 0.14  | 0.41$\pm$ 0.17 |
> | CEM+compatibility × \|M\| | **0.23** $\pm$ 0.32 | **0.43** $\pm$ 0.34 | **0.14** $\pm$ 0.22 | **0.57** $\pm$ 0.3 | **0.66** $\pm$ 0.08 |
>
> The results show that, across every task listed above, CEM+compatibility × |M| outperforms CEM × |M|, demonstrating the effectiveness of employing the proposed compatibility measure.
>
> We thank the reviewer for inspiring us to conduct this experiment, and we will include the results in the revised paper.
>
> >While generating reusable subroutines is intuitive, there are no examples on how these low-level policies look like throughout the paper, making it a bit harder to concretely visualize how the policies look like. Adding a few examples, even in appendix, would make the process more interpretable.
>
> We provide multiple examples in Figures 21, 22, 23, 24 in the appendix. These examples serve as concrete examples that explain what the agent will do by following these programmatic options. Taking Inf-DoorKey as an example, this task requires iterating between three different skills.
> - The first skill is picking up the key in a chamber to open the door. Option 1 of this task, shown in Figure 23, is suitable for acquiring this skill since this program will check whether a key exists in the agent position and pick it up to open the door.
> - The second skill is navigating inside and between chambers. Option 2 of this task, shown in Figure 23, is suitable for acquiring this skill since this program contains numerous tokens about moving and turning directions.
> - The third skill is locating and placing the key in a specific position to open the door. Option 3 of this task, shown in Figure 23, is suitable for acquiring this skill since this program will first place the key and then pick it up.
>
> By combining these skills with the help of the high-level policy controller, the agent can pick up or place keys in the correct places and navigate inside and between chambers.
>
> >Does the choice of DSL affect the performance of the proposed process? Would a less well-designed DSL cause the method to be ineffective?
>
> As pointed out by the reviewer, the choice of DSL plays a vital role in the proposed framework. In this work, we choose Karel DSL because its semantics are designed to describe an agent’s behavior and how agents can observe and interact with the environment. Suboptimal DSLs (e.g., inaccurate tokens to indicate state or behavior) could lead to suboptimal programmatic policies. We will incorporate this discussion into the limitation section (Section J) in the paper revision.

---

> > ### Comment · Reviewer_4qWN · 2024-08-12
> >
> > Thank the authors for the detailed response and additional experiments. The discussions and potential adjustments sound reasonable to me. I will keep my score.

---

### Official Review · Reviewer_p5Z4 · 2024-07-10

**Soundness:** 3
**Presentation:** 3
**Contribution:** 2
**Rating:** 5
**Confidence:** 3

**Summary:**

Utilizing human-readable programs as policies has been recently proposed to enhance interpretability in reinforcement learning. This work introduces the Hierarchical Programmatic Option Framework (HiPO) that first embeds the programs into a smooth and continuously parametrized space, obtains a diverse and compatible skill set by applying the cross entropy method on the embedding space with respect to novel diversity and compatibility metrics, and finally learning a high-level policy upon the skill set. Experimental analysis of HiPO on the Karel problem sets manifests its effectiveness and zero-shot generalizability.

**Strengths:**

The authors proposed novel reward functions that can enhance the diversity and compatibility of the program skill set obtained by CEM. The experimental results show that HiPO performs better on average compared to existing baselines, and every part of HiPO plays a significant role.

**Weaknesses:**

1. The execution time will differ from program to program, but HiPO does not consider this factor while incorporating a discount factor $\gamma$ of 0.99. The adoption of SMDP should be considered to deal with the variance in execution time appropriately. Also, HiPO does not discount the cumulative reward of low-level actions, further deepening the inconsistency between the theoretical objective and the actual loss function. The authors should justify such design choices.

2. In Section 5.3, the difference between the two settings, CEM+diversity top k and CEM+diversity x |M|, is unclear. Further explanation would be helpful for the readers.

3. The authors did not conduct experiments on the CEM + compatibility setting. In the Karel-Long environments, most of the improvement seems to come from the compatibility heuristic.

**Questions:**

1. Figure 6 shows how the adoption of the diversity factor spreads the skill set over the embedding space. However, the diversity of the embedding vectors does not necessarily guarantee the diversity in the actual behaviour. How different are the programs' actual behaviors in the final skill set?

2. From Section A.2, it seems like at every step, the procedure selects one program through CEM. Wouldn't this approach cause the programs selected in the earlier stages to be sub-optimal?

**Limitations:**

The authors adequately addressed the limitations of their work.

---

> ### Author Rebuttal · Authors · 2024-08-06
>
> We sincerely thank the reviewer for the thorough and constructive comments. Please find the response to your questions below.
>
> >The execution time will differ from program to program, but HiPO does not consider this factor while incorporating a discount factor 𝛾 of 0.99.
>
> During the option retrieval process, the normalized reward defined in Equation 1 explicitly considers the execution time of each program in the sampled option sequence through the discount factor 𝛾 (i.e., distant reward will be discounted more than instant reward in each program execution). Therefore, the evaluation function $G$ introduced in Section 4.2.3 not only assesses the diversity and compatibility among different options, but also considers the variance of execution time of each option across lists of options sampled by the evaluation function. We will revise the paper to make it clear.
>
> >HiPO does not discount the cumulative reward of low-level actions
>
> After retrieving a set of options, each option can be viewed as a “macro action” by the high-level policy, and executing each macro action takes a “macro timestep” (i.e., the beginning of the episode or after fully executing the selected option), as depicted in Figure 2(b). Since these options are fixed and deterministic sub-policies in sMDP, we use the cumulative reward of all the low-level actions from a program execution as the return of each macro action. This aligns with existing hierarchical RL literature [1,2,3].
>
> >The difference between the two settings, CEM+diversity top k and CEM+diversity x |M|, is unclear.
>
> In CEM+diversity top k (k=|M|), we select the top k=|M| programs out of a total of N=10 CEM searches. On the other hand, CEM+diversity x |M| conducts |M| rounds of N=10 CEM search (a total of 10 x |M| CEM searches) and selects the best program in each round to get a total of |M| programs.
>
> Note that when conducting the j-th CEM during the i-th CEM+diversity of CEM+diversity x |M|, the diversity multiplier is calculated based on the previous (i-1) retrieved programs and the current j programs. In comparison, the j-th CEM during CEM+diversity top k only considers the diversity among these j programs. We will revise the paper to make it clear.
>
> >The authors did not conduct experiments on the CEM + compatibility setting.
>
> As suggested by the reviewer, we conduct additional experiments of CEM+compatibility ×|M|.
>
> | Method | Seesaw | UP-N-Down | Farmer | Inf-DoorKey | Inf-Harvester |
> |-|-|-|-|-|-|
> | CEM × \|M\| | 0.06 $\pm$ 0.10 | 0.39 $\pm$ 0.36 | 0.03 $\pm$ 0.00 | 0.11 $\pm$ 0.14  | 0.41$\pm$ 0.17 |
> | CEM+diversity top k, k=\|M\|| 0.15 $\pm$ 0.21 | 0.25 $\pm$ 0.35 | 0.03 $\pm$ 0.00 | 0.13 $\pm$ 0.16 | 0.42$\pm$ 0.19 |
> | CEM+diversity × \|M\|| **0.28** $\pm$ 0.23 | **0.58** $\pm$ 0.31 | 0.03 $\pm$ 0.00 | 0.36 $\pm$ 0.26 | 0.47$\pm$ 0.23 |
> | CEM+compatibility × \|M\| | 0.23 $\pm$ 0.32 | 0.43 $\pm$ 0.34 | **0.14** $\pm$ 0.22 | **0.57** $\pm$ 0.3 | **0.66** $\pm$ 0.08 |
>
> Across tasks listed above, CEM+compatibility × |M| performs better than CEM × |M|, showing the effectiveness of the compatibility measure on its own. For multi-stage tasks like Farmer and Inf-DoorKey, CEM+compatibility × |M| performs better than CEM+diversity top $k$ and CEM+diversity × |M|, indicating that the compatibility measure can help CEM more easily find skills suitable for multi-stages compared to the diversity measure. We will revise the paper to include this experiment.
>
> >The diversity of the embedding vectors does not necessarily guarantee the diversity in the actual behaviour.
>
> We learn a program embedding space following Trivedi et al. [4]. The objective function features a latent behavior reconstruction loss, which aims to ensure that in the learned embedding space, similar behaviors are encoded closer and drastically different behaviors are encoded far from each other.
>
> >How different are the programs' actual behaviors in the final skill set?
>
> Figures 21, 22, 23, and 24 present the retrieved options using our method. Taking Inf-DoorKey as an example, this task requires iterating between three skills:
> - Picking up the key in a chamber to open the door. Option 1 in Figure 23 is suitable for acquiring the skill since this program will check whether a key exists in the agent position and pick it up to open the door.
> - Navigating inside and between chambers. Option 2 in Figure 23 is suitable for acquiring the skill since this program contains numerous tokens about moving and turning directions.
> - Locating and placing the key in a specific position to open the door. Option 3 in Figure 23 is suitable for acquiring the skill since this program will first place the key and then pick it up.
>
> >From Section A.2, it seems like at every step, the procedure selects one program through CEM. Wouldn't this approach cause the programs selected in the earlier stages to be sub-optimal?
>
> The programs retrieved in the earlier stages could be suboptimal, which motivates us to design the diversity and compatibility rewards to ensure the performance of the retrieved programs as a whole. Despite the potential suboptimal retrieved programs, the experimental results in Sections 5.2, 5.3, and 5.4 show that the high-level policy can effectively reuse programmatic options selected by the proposed procedure, outperforming the baselines.
>
> ---
>
> References:
>
> [1] Ofir Nachum, Shixiang Gu, Honglak Lee, and Sergey Levine. Data-efficient hierarchical reinforcement learning. In NeurIPS, 2018.
>
> [2] Kevin Frans, Jonathan Ho, Xi Chen, Pieter Abbeel, and John Schulman. Meta learning shared hierarchies. In ICLR, 2018.
>
> [3] Youngwoon Lee, Jingyun Yang, and Joseph J. Lim. Learning to coordinate manipulation skills via skill behavior diversification. In ICLR, 2020.
>
> [4] Dweep Trivedi, Jesse Zhang, Shao-Hua Sun, and Joseph J Lim. Learning to synthesize programs as interpretable and generalizable policies. In NeurIPS, 2021.

---

> ### Author Response · Authors · 2024-08-10
> **Reminder: The reviewer-author discussion period ends in three days**
>
> We would like to express our sincere gratitude to the reviewer for the thorough and constructive feedback. We are confident that our responses adequately address the concerns raised by the reviewer, including the following points.
> - A discussion of discount factors in our hierarchical RL setup
> - A clarification of the two settings, CEM+diversity top k and CEM+diversity x |M|
> - Additional results of CEM + compatibility
> - An explanation of diversity in program embeddings and behaviors
> - A discussion of suboptimal retrieved programs
>
> Please kindly let us know if the reviewer has any additional concerns or if further experimental results are required. We are fully committed to resolving any potential issues, should time permit. Again, we thank the reviewer for all the detailed review and the time the reviewer put into helping us to improve our submission.

---

> > ### Comment · Reviewer_p5Z4 · 2024-08-12
> >
> > I'm very sorry for the late reply. Here are some clarifications I want to make.
> >
> > Unlike [1, 2, 3], where the length of a high-level action is fixed, HIPO deals with variable-length high-level actions. This can be problematic in certain situations. Consider the following two sequences of rewards:
> >
> >     1) (0, 1, 0, 1, 0, 1), (0, 1)
> >     2) (0, 1), (0, 1), (0, 1), (0, 1.01)
> >
> > where rewards from the same macro-action are grouped by parentheses. Under the return computation scheme used by HIPO, sequence 1 results in $4+0.99\times 1=4.99$ and sequence 2 results in $1+0.99\times 1+0.99^2\times 1+0+0.99^3\times 1.01\approx 3.95$. Even though sequence 2 is better, HIPO will prefer the first one.

---

> > > ### Author Response · Authors · 2024-08-13
> > > **Re: Official Comment by Reviewer p5Z4**
> > >
> > > We thank the reviewer for the question with further clarification.
> > >
> > > > Unlike [1, 2, 3], where the length of a high-level action is fixed, HIPO deals with variable-length high-level actions. This can be problematic in certain situations. Consider the following two sequences of rewards ...
> > >
> > > Due to the temporal abstraction brought by the hierarchical structure, the high-level policy evaluates the effectiveness (i.e., cumulative reward) of each macro action no matter how long or short each macro action is. That said, we consider learning the high-level policy to be a standard RL problem. As pointed out by the reviewer, with a discount factor $\gamma<1$, the high-level policy prefers immediate rewards to delayed rewards. This has been proven to be effective and lead to better convergence, and is widely adopted in most RL algorithms [1-4] with or without a hierarchical policy structure. Note that such a setting is also adopted by [5], whose low-level policy (transition policy) also has varying horizons.
> > >
> > > On the other hand, the preference of the high-level policy can be adjusted under different values of $\gamma$. For example, if the value of $\gamma$ is raised from $0.99$ to $0.999$, the discounted return of sequence 1 in the example above is $3 + 0.999 \times 1 = 3.999$, and the discounted return of sequence 2 is $1 + 0.999 \times 1 + 0.999^2 \times 1 + 0.999^3 \times 1.01 \approx 4.003974$. With this adjustment, the high-level policy prefers sequence 2. Hence, the desired value of the discount factor can vary from one task to another.
> > >
> > > We thank the reviewer for the fruitful discussion and will revise the paper to include it. Also, we hope our initial rebuttal sufficiently addresses other questions raised in your initial review. Please kindly let us know if the reviewer has any additional concerns or if further experimental results are required. We are fully committed to resolving any potential issues, should time permit.
> > >
> > > [1] Nachum et al., "Data-efficient hierarchical reinforcement learning." In NeurIPS, 2018.
> > >
> > > [2] Frans et al., "Meta learning shared hierarchies." In ICLR, 2018.
> > >
> > > [3] Lee et al., "Learning to coordinate manipulation skills via skill behavior diversification." In ICLR, 2020.
> > >
> > > [4] Schulman et al., "Proximal policy optimization algorithms." 2017.
> > >
> > > [5] Lee et al., "Composing Complex Skills by Learning Transition Policies." In ICLR, 2019.

---

> > > > ### Comment · Reviewer_p5Z4 · 2024-08-14
> > > >
> > > > My point was to suggest discounting more after long macro-actions, e.g. $(\gamma+\gamma^3+\gamma^5)+\gamma^6\times\gamma$ for the first sequence and $\gamma+\gamma^2\times\gamma+\gamma^4\times\gamma+\gamma^6\times 1.01\gamma$ for the second one. Anyway, my other concerns are resolved, and I have updated my score. Please add the explanations to the camera-ready version.

---

### Official Review · Reviewer_ukMP · 2024-07-12

**Soundness:** 4
**Presentation:** 4
**Contribution:** 3
**Rating:** 7
**Confidence:** 4

**Summary:**

The authors present HIPO, a method that use a Program embedding space to create options. Some are retrieve if diverse and efficient to create a set of option, later used by a learned high level policy. To evaluate their framework, as the KAREL benchmark does not include long and repetitive tasks, they introduce KAREL-LONG and evaluate their method on both benchmarks. Their result show that HIPO is a viable hierarchical RL method.

**Strengths:**

The paper is sound and clear. The structure is clear, the contributions easily identifiable, the figures, table are easily understandable.
I am not a deep expert of option or program synthesis, but the paper is so clear that it brought me sufficient insight to situate it and understand its method. The KAREL-LONG test-bench is well motivated, as one can see that the original framework is not enough to differentiate its existing SOTA methods.
Their evaluation shows that HIPO helps with long and repetitive tasks, while being the strongest competitor on the existing KAREL domain.
I'm already giving an accept, but might further raise based on the other reviews and the rebuttal.

**Weaknesses:**

The only main weakness I was able to identify is the:

**Lack of limitation section**. A dedicated *Limitation* section could clearly help situate the advantages and drawback of HIPO, and compare it to other methods. Authors could step a little bit outside programmatic and look into e.g. interpretable logic-based RL [1, 2, 9, 10], interpretable RL methods that use LLM [3], which can be useful for programs synthesis, tree-based policies (convertible to tree programs)[8, 5, 6, 7], ... etc. I think the "background" part of current related work section could be included in the *Introduction*, and that a dedicated *Limitation and related work* section could include such a discussion. It would situate the method in the broader literature, and bring attention to the nice HIPO method to these communities as well. I give some pointers references, but some more could be searched.

Further smaller concerns are:

**Interpretability allows to detect invisible suboptimal behaviors.** It has been shown that non-interpretable (e.g. deep) RL methods often learn misaligned policies [4, 5, 6], without this problem being spotted, on tasks as simple as Pong [5]. This is a much stronger argument to directly place in the introduction than the lack of transparency and trust. I would start off using such an argument, highlighting the need of e.g. readable, interpretable programs.

**Authors could slightly modify your method name (if they wish).** Your methods' name might collide with *Hierarchical Proximal Policy Optimization (HIPPO)*, an existing known option learning framework.

-------------------------------
[1] Jiang, Zhengyao, and Shan Luo. "Neural logic reinforcement learning." International conference on machine learning. PMLR, 2019.

[2] Xu, Duo, and Faramarz Fekri. "Interpretable model-based hierarchical reinforcement learning using inductive logic programming." arXiv preprint arXiv:2106.11417 (2021).

[3] Luo, Lirui, et al. "INSIGHT: End-to-End Neuro-Symbolic Visual Reinforcement Learning with Language Explanations." arXiv preprint arXiv:2403.12451 (2024).

[4] Di Langosco, Lauro Langosco, et al. "Goal misgeneralization in deep reinforcement learning." International Conference on Machine Learning. PMLR, 2022.

[5] Delfosse, Quentin, et al. "Interpretable concept bottlenecks to align reinforcement learning agents." arXiv preprint arXiv:2401.05821 (2024).

[6] Kohler, Hector, et al. "Interpretable and Editable Programmatic Tree Policies for Reinforcement Learning." arXiv preprint arXiv:2405.14956 (2024).

[7] Delfosse, Quentin, et al. "HackAtari: Atari Learning Environments for Robust and Continual Reinforcement Learning." arXiv preprint arXiv:2406.03997 (2024).

[8] Bastani, Osbert, Yewen Pu, and Armando Solar-Lezama. "Verifiable reinforcement learning via policy extraction." Advances in neural information processing systems 31 (2018).

[9] Delfosse, Quentin, et al. "Interpretable and explainable logical policies via neurally guided symbolic abstraction." Advances in Neural Information Processing Systems 36 (2024).

[10] Ma, Z., Zhuang, Y., Weng, P., Zhuo, H. H., Li, D., Liu, W., & Hao, J. (2021). Learning symbolic rules for interpretable deep reinforcement learning. arXiv preprint arXiv:2103.08228.

**Questions:**

* Are the KAREL-LONG environment containing training and testing environments ? As programs are compared, it might be insightful to test the generalizability of the different methods (by varying size, of the environment).
* I am not sure if the option is returning a terminal token or if the high level controller is reselecting an option at each timestep. This could be made clearer.
* What's the percentage of retrieved program (based on your effectiveness and diversity criterion) ?

**Limitations:**

This constitutes my main concern, I have not been able to clearly situate a (broader discussion) on the limitations. Again, I would create such a section, bringing most of the related work in it, as an opportunity to criticize (both positively and negatively) HIPO, and other approaches.

---

> ### Author Rebuttal · Authors · 2024-08-06
>
> We sincerely thank the reviewer for the thorough and constructive comments. Please find the response to your questions below.
>
> >A dedicated Limitation section could clearly help situate the advantages and drawback of HIPO, and compare it to other methods. Authors could step a little bit outside programmatic and look into e.g. interpretable logic-based RL [1, 2, 9, 10], interpretable RL methods that use LLM [3], tree-based policies [8, 5, 6, 7], ... etc.
>
> We thank the reviewer for providing the references and suggestions. We will revise the paper to discuss these works by adding a dedicated section or merging it with Section J, which points out the challenges of DSL design and the interpretability-performance tradeoff.
>
>
> >Interpretability allows to detect invisible suboptimal behaviors
>
> We thank the reviewer for this insight. We totally agree that interpretability can help detect misaligned policies. We will revise the paper to incorporate this discussion.
>
> >Authors could slightly modify your method name (if they wish). Your methods' name might collide with Hierarchical Proximal Policy Optimization (HIPPO), an existing known option learning framework.
>
> We thank the reviewer for the suggestion. We will try to adjust the name of our method to avoid potential confusion.
>
> >It might be insightful to test the generalizability of the different methods (by varying size, of the environment).
>
> In Section 5.6, we evaluate the ability to inductive generalization in testing Karel-Long environments with longer horizons than training environments. Table 4(b) indicates that HIPO generalizes better to the testing environments with significantly extended horizons compared to the baselines.
>
> We will add other different generalization settings (e.g., varying size of the environment) to further testify the proposed framework in the  revision.
>
> >I am not sure if the option is returning a terminal token or if the high level controller is reselecting an option at each timestep.
>
> As detailed in Figure 2 and Section 4.3, the high-level controller will reselect an option after the previously selected programmatic option is executed and terminated. After executing the selected programmatic option, the high-level policy will receive the execution trace (i.e., a series of (state, action, reward) tuples) collected during the program execution. Based on the final state in the execution trace and the current selected programmatic option, the high-level policy will choose the next programmatic option. We will revise the paper to make this clear.
>
> >What's the percentage of retrieved program (based on your effectiveness and diversity criterion)?
>
> The theoretical search space of each programmatic option is about $30^{40}$, i.e., the token space size to the power of program length. Throughout the experiments conducted in this paper, we sample approximately 50000 unique programs according to our effectiveness and diversity criteria to solve a given task, which only takes up a small fraction of the total program space; therefore, an effective retrieval method is essential.

---

> > ### Comment · Reviewer_ukMP · 2024-08-11
> > **Thank you for the clarifications**
> >
> > I want to thank the authors for their clarifications.
> > I want to insist on the fact that they do not have to change their method name, I just gave an advice, but the choice is theirs.
> >
> > If the last point above has not been discussed in the paper, I would advise adding it as well.
> >
> > Overall, I hope that the promised modifications will be incorporated in the manuscript, and hope to see this paper presented at the conference.

---

> > > ### Author Response · Authors · 2024-08-11
> > > **Re: Thank you for the clarifications**
> > >
> > > We would like to express our sincere gratitude to the reviewer for the thorough and constructive feedback. We will definitely revise the paper to incorporate all the modifications that were promised.

---

### Official Review · Reviewer_x8GU · 2024-07-14

**Soundness:** 2
**Presentation:** 3
**Contribution:** 3
**Rating:** 7
**Confidence:** 5

**Summary:**

The paper presents HIPO, a method for learning programmatic options for solving problems with long-horizon and repetitive tasks. First, HIPO searches in the space defined by a domain-specific language for a set of diverse programs. This set of programs is generated while accounting for the diversity and composability of the programs. Finally, these programs are used as options for a neural policy. HIPO was evaluated on instances of Karel the Robot, where it was shown to be more sample efficient than Deep RL and other programmatic methods. The paper also includes a carefully designed ablation study on HIPO's components.

**Strengths:**

The idea of using programmatic representations to learn options is interesting and valuable. The authors correctly state that using programmatic policies as options as opposed to policies means that one sacrifices, to some extent, the interpretability of the policies in favor of performance.

Another strength of HIPO is its natural combination of programmatic and neural representations -- this is not mentioned in the paper and perhaps it should! The repetitive behavior required to solve the tasks is given by the programmatic options, which can achieve this through the strong inductive bias the Karel language offers. The neural policy is then responsible for orchestrating these different components, which would clearly be difficult through program synthesis.

I am particularly impressed by the results on the DoorKey domain. This is such a difficult domain for program synthesis and DRL. HIPO managed to achieve a really high score in that domain. I do have some clarifying questions about this result (please see them in the Questions section of this review).

I also appreciated the extra effort the authors put into translating the neural model that orchestrates the options into automata for interpretability. Since the automata can also be seen as programs, it would be interesting to see their performance or even their translation into the Karel language.

**Weaknesses:**

The paper has a few weaknesses too. It would be helpful if the authors managed to fix the following for the camera ready, if the paper is accepted.

1. HIPO represents a method for learning programmatic options, but it includes no option-learning baseline.
2. Searching directly in the programmatic space was recently shown to outperform the latent search used in HIPO. It would be valuable to include this type of search as a baseline too.
3. It would be interesting to measure sample efficiency in terms of samples instead of programs. Intuitively, some programs could run much longer and be more costly in real settings. This would also allow for a comparison with DRL in terms of sample complexity. In general, I find learning curves more informative than the tables presented in the paper, which show only asymptotic performance.
4. It would be valuable to have more runs of the experiments >30 and show confidence intervals instead of standard deviation, as CI is easier to draw conclusions from without statistical tests.

Overall, in my opinion, the strengths of the paper outweigh its weaknesses. This is because the paper shows a different way of using programmatic representations that the community should further explore.

**Questions:**

Please feel free to comment on the points listed under weaknesses. In addition to them, I will add the following questions and comments.

1. I am puzzled by the outstanding performance of HIPO in DoorKey. The way that the options are selected is that they have to be diverse and they have to maximize the agent's return. Since the options are selected as a pre-processing step, before orchestrating them, the programs must be collecting the key and receiving a positive reward, all of them. Once they are orchestrated, they happen to contain the behavior needed to find the door and collect the marker after the key is found. Is it because the composability step is already finding a solution that stitches together options that are able to solve the problem? If that is the case, would it be valuable to add a baseline that performs diversity+composability and returns the best order used in the composability step?

2. Why does HIPO use an option that terminates the episode? This doesn't make sense to me. The episode finishes whenever it finishes. That is, it is not up to the agent to decide when the episode finishes. This is a property of the MDP.

3. The caption of Figure 3 says that the Harvester goes from 36 to 400 markers as we move from Harvester to Inf-Harvester. Due to the structure of the problem, isn't it possible to find a short program that is able to collect an arbitrary number of markers?

4. I would like to challenge the desiderata of the options (lines 136-139). First, an option doesn't have to be effective in the sense that it needs to "obtain some task reward." It just needs to be part of the solution. For example, in the DoorKey problem, if the agent only received a positive reward once the problem was solved, an option that collects the key would still be helpful. Second, I suspect that the property of compatibility is only needed because the neural agent doesn't have access to the primitive actions while orchestrating the options. If the agent had access to them, it would be able to combine any "helpful option" by using primitive actions in between them. However, I understand that even if the agent had access to primitive actions, it could still find value in compatibility. In this case, compatibility would be playing the effectiveness property, and it would be looking at ways of how different programs can be combined to maximize the agent's return.

5. How can the diversity score be defined for searching directly in the programmatic space? Could one use the losses used to train the latent space to define diversity? Or is the diversity score only possible because of the properties of the latent space?

**Limitations:**

The paper has a limitations section in the appendix that addresses some important limitations, such as the trade-off between interpretability and performance. I would also add the points raised in the Weaknesses section of this review and the use of a single domain in the experiments.

---

> ### Author Rebuttal · Authors · 2024-08-07
>
> We sincerely thank the reviewer for the thorough and constructive comments. Please find the response to your questions below.
>
> > Option-learning baselines
>
> As suggested by the reviewer, we additionally experimented with the option-critic architecture [1] and reported the comparison to our method below.
>
> | Method | Seesaw | UP-N-Down | Farmer | Inf-DoorKey | Inf-Harvester |
> |-|-|-|-|-|-|
> | Option-critic | 0.00 $\pm$ 0.00 | 0.00 $\pm$ 0.00 | 0.00 $\pm$ 0.00 | 0.00 $\pm$ 0.00 | 0.47 $\pm$ 0.01 |
> | HIPO (Ours) | **0.53** $\pm$ 0.10 | **0.76** $\pm$ 0.02 | **0.62** $\pm$ 0.02 | **0.66** $\pm$ 0.07 | **0.79** $\pm$ 0.02 |
>
> The results show that our method outperforms option-critic on all the Karel-Long tasks. Option-critic performs poorly on all tasks except for Inf-Harvester, likely because of the sparse rewards nature of these tasks and the environment's per-action cost, possibly forcing the options to be trained to terminate quickly. We will revise the paper to include this result.
>
> > Search in the programmatic space
>
> As suggested by the reviewer, we conduct additional experiments that search directly in the programmatic space using the hill climbing (HC) approach proposed by Carvalho et al. [2].
>
> | Method | Seesaw | UP-N-Down | Farmer | Inf-DoorKey | Inf-Harvester |
> |-|-|-|-|-|-|
> | HC | 0.22 $\pm$ 0.08 | 0.63 $\pm$ 0.26 | 0.19 $\pm$ 0.03 | 0.14 $\pm$ 0.16 | **0.88** $\pm$ 0.00 |
> | HIPO (Ours) | **0.53** $\pm$ 0.10 | **0.76** $\pm$ 0.02 | **0.62** $\pm$ 0.02 | **0.66** $\pm$ 0.07 | 0.79 $\pm$ 0.02 |
>
> The results show that searching directly in the programmatic space can achieve better performance on tasks with denser rewards, e.g., Inf-Harvester. On the other hand, HIPO performs better on sparse-reward tasks requiring diverse skills. We will revise the paper to include this baseline.
>
> > Environment-step sample efficiency
>
> We thank the reviewer for the suggestion. We will add this plot to the revised paper.
>
> > More runs of the experiments >30 and confidence intervals
>
> We thank the reviewer for the suggestion. We presented standard deviations with five random seeds following the standard practice in RL literature. We will add more runs and show confidence intervals in the revision.
>
> > Performance of HIPO in DoorKey
>
> The diversity multiplier encourages exploring programs of different behaviors during the search process. Therefore, after retrieving the program that "find the key" as the first option, other behaviors like "navigating" or "put maker" will be more likely to be retrieved as the second option.
>
> Due to the consideration among the options, it is possible to sample some option execution orders that are able to solve the problem following the evaluation function defined in Section 4.2.3. If one program can be stitched together with retrieved options in some specific order to solve the program, then the evaluation function will return a high score for this new option.
>
> > Best order used in the composability step
>
> As suggested by the reviewer, we report the return obtained by executing the best random sequence order found in the compatibility step during the search of CEM+diversity+compatibility.
>
> | Method | DoorKey | Seesaw | UP-N-Down | Farmer | Inf-DoorKey | Inf-Harvester |
> |-|-|-|-|-|-|-|
> | Best Random Sequence | **1.00** $\pm$ 0.00 | 0.04 $\pm$ 0.02 | 0.17 $\pm$ 0.08 | 0.05 $\pm$ 0.03 | 0.06 $\pm$ 0.06 | 0.60 $\pm$ 0.02 |
> | HIPO (Ours) | **1.00** $\pm$ 0.00 | **0.53** $\pm$ 0.10 | **0.76** $\pm$ 0.02 | **0.62** $\pm$ 0.02 | **0.66** $\pm$ 0.07 | **0.79** $\pm$ 0.02 |
>
> The results show that, without the high-level policy, executing options with a specific sequence can already achieve a good performance on tasks with short-horizon or denser reward tasks requiring fewer skills, e.g., DoorKey and Inf-Harvester. On the other hand, HIPO performs better on long-horizon and sparse-reward tasks, indicating the necessity of integrating the high-level policy with the option searching process when solving repetitive and long tasks.
>
> > Termination option
>
> We implement the termination option as a “do nothing” low-level policy, i.e., an empty program, that triggers null action to avoid the per-action cost as described in Section H, which urges the agent to solve the tasks as efficiently as possible by reducing redundant actions. That said, as pointed out by the reviewer, the MDP decides when to terminate, and the agent can learn to take null action to avoid action costs.
>
> > A short program that is able to collect an arbitrary number of markers
>
> Yes, it is possible to find a short program that collects an arbitrary number of markers in the Inf-Harvester task. Despite the long-horizon nature of this task, it only requires repeatedly traversing and picking markers, which can be easily encapsulated by short programs with a small number of primitive actions inside an outer WHILE control statement.
>
> > I would like to challenge the desiderata of the options (lines 136-139).
>
> We thank the reviewer for the insight. We will remove “(i.e., obtain some task rewards)” from the definition of effectiveness.
>
> > Diversity score for programmatic spaces
>
> We think it is not trivial to define the diversity score in a programmatic space. Specifically, taking the programmatic space defined by abstract syntax trees (ASTs) proposed in Carvalho et al. [2] as an example, it is not clear how we can compute the distance between a pair of ASTs. Exploring how the diversity score can be defined in programmatic spaces is an interesting future direction.
>
> ---
>
> References:
>
> [1] Pierre-Luc Bacon, Jean Harb, and Doina Precup. The option-critic architecture. In Association for the Advancement of Artificial Intelligence, 2017.
>
> [2] Tales Henrique Carvalho, Kenneth Tjhia, and Levi Lelis. Reclaiming the source of programmatic policies: Programmatic versus latent spaces. In ICLR, 2024.

---

> > ### Comment · Reviewer_x8GU · 2024-08-13
> >
> > Thank you for answering all my questions, I appreciate the effort.

---

### Decision · Program_Chairs · 2024-09-25

**Decision:**

Accept (poster)

**Comment:**

This paper presents a method for learning options as programs for repetitive, long-horizon tasks. We all agreed the work is well-written and clear, the method well-motivated and well-defined, and the empirical evaluations convincing enough to recommend an acceptance.